# CHAININFER: A JOINT METHOD FOR INFERRING MISSING AI SUPPLY CHAIN INFORMATION

## ABSTRACT

The modern AI ecosystem forms an intricate *AI supply chain*, where models, datasets, and software components are deeply interconnected. Incomplete or inconsistent metadata on platforms such as Hugging Face and Kaggle leaves critical gaps in provenance, hindering reproducibility, risk management, and governance. To address this, we formalize **AI supply chain inference** as a coupled graph learning problem: *link prediction to recover missing dependencies* and *edge classification to determine their semantic types*. We propose CHAININFER, a hybrid architecture that integrates graph neural networks for local structural reasoning with graph transformers for global context, trained end-to-end on attributed supply chain graphs. Using a benchmark of 200K models from Hugging Face, CHAININFER outperforms GNN-, Transformer-, and ensemble baselines, achieving 0.93 joint accuracy while remaining efficient. Moreover, CHAININFER generalizes inductively to Kaggle, retaining 0.90 accuracy without retraining. These results demonstrate CHAININFER as a practical framework for scalable, accurate, and transferable AI supply chain provenance inference.

## 1 INTRODUCTION

The modern Artificial Intelligence (AI) ecosystem no longer consists of isolated models or datasets. Instead, it resembles a complex, interdependent *AI supply chain* in which foundation models, fine-tuned derivatives, datasets, software stacks, and specialized hardware continually depend on and influence one another (Wang et al., 2025; Bommasani et al., 2024). The term AI supply chain (NIST, 2023b; Huang et al., 2024) denotes this interconnected pipeline of components that collectively enable the development, deployment, and maintenance of AI systems. Figure 1 denotes an AI supply chain example centering on the Meta-LLaMa model (Touvron et al., 2023).

As AI permeates critical applications and infrastructure, understanding and managing the AI supply chain is essential for four reasons. (i) *Hidden dependencies* allow a single flaw (e.g., a vulnerable model, a poisoned dataset, or a licensing conflict) (Carlini et al., 2021; Lehman et al., 2021; Kandpal et al., 2022; White et al., 2023; Zhang et al., 2023) to propagate silently to large numbers of downstream systems. For example, when *LLaMA-2-Chat* exhibited jailbreak-induced unsafe outputs (Chen et al., 2023a), many fine-tuned variants such as Vicuna (Chiang et al., 2023), Alpaca (Taori et al., 2023), and Koala (Geng et al., 2023) were observed to inherit

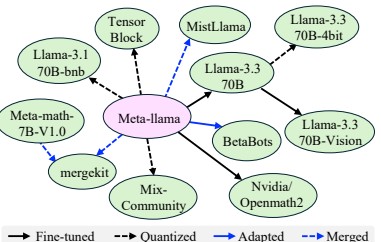

Figure 1: An AI supply chain example centering on Meta-LLaMA.

similar issues (Zou et al., 2023; Shi et al., 2024; He et al., 2024). (ii) *Transparent provenance* is increasingly mandated for reproducibility and accountability (e.g., the EU AI Act) (EUA, 2024). (iii) *Risk management and governance* require visibility into systemic failure points to enforce reliable practices (e.g., NIST AI RMF) (NIST, 2023a). (iv) *Supply chain awareness* improves efficiency and innovation by revealing redundant effort and opportunities for reuse, including gaps in coverage (e.g., under-served languages/modalities) that guide resource allocation (Taraghi et al., 2024).

The AI platforms such as Hugging Face (Hugging Face) and Kaggle (Kaggle Inc., 2025) are emerging as the major vendors in AI model management and sharing. The model information is typically provided through metadata (e.g., model cards) describing attributes, dependencies, and architecture.

However, developer-supplied metadata lacks consistent standards and is often *missing* or *incomplete*. Particularly, a recent work reported that, more than half models on Hugging Face (as of June 2025) miss or have incomplete metadata (Rahman et al., 2025).

To compensate for incomplete metadata, prior efforts infer lineage using content- or behavior-level signals. Similarity and fingerprinting methods compare architectures, parameters, and functional behavior (e.g., Model2Vec (Tulkens & van Dongen, 2024), DeepJudge (Chen et al., 2023b), Tensorguard (Wu et al., 2025)), while difference-based and representation-similarity methods (e.g., ModelDiff (Shah et al., 2023), CKA (Kornblith et al., 2019)) analyze weights or activations to identify reuse. Although these techniques can recover explicit edges when metadata is trustworthy, they often depend on deterministic heuristics or static parsing and struggle to generalize, scale to one (or many)-to-many dependencies, or capture latent obscured relationships across millions of artifacts.

We observe, despite being partial, model cards and repository metadata frequently contain *complementary* cues (e.g., declared dependencies, and architecture summaries). When aggregated, they would form an attributed graph revealing both explicit and implicit relationships. *This motivates casting **AI supply chain inference** as two coupled predictive tasks on a partially observed graph: **link prediction to identify missing edges** and **edge classification to determine the types***.

Recent advances in graph learning, such as graph neural networks (GNNs) (Kipf & Welling, 2016) and graph transformers (GTs) (Dwivedi & Bresson, 2020; Ying et al., 2021a), are natural methods for these tasks. Message-passing GNNs efficiently capture local structure but might suffer from over-smoothing/over-squashing, limiting multi-hop reasoning in deep supply chains. GTs model long-range dependencies via global attention while they might be low efficient due to global attention. Moreover, most existing approaches (Jin et al.; Diao & Loynd; Wu et al., 2023) are optimized for either link prediction or edge classification in isolation, leaving performance on the coupled problem suboptimal.

Motivated by that, we devise CHAININFER, a new joint method for inferring missing AI supply chain information. Particularly, we make the following three major contributions.

- **Problem formulation.** First, we formalize AI supply chain inference on an attributed, typed, and directed graph built from public repositories explicitly addressing incomplete, and inconsistent metadata by coupling *link prediction* with *edge classification*.
- **Hybrid architecture.** Second, we propose a joint model that fuses GNN layers for local structural signals with GT layers for global context. Trained end-to-end on the same attributed graph, the fused representation supports both tasks simultaneously, improving accuracy while mitigating the computational overhead of full-attention transformers.
- **Extensive evaluation.** Lastly, we construct a large-scale benchmark for AI supply chain inference from Hugging Face and conduct extensive experiments and ablations. Particularly, CHAININFER achieves 0.93 accuracy (joint edge classification and link prediction), significantly higher than the best of separately trained models (0.89), GT-only methods (0.86), and GNN-only methods (0.75). In addition, CHAININFER retains 0.90 joint accuracy when transferred inductively to another AI supply chain graph collected from Kaggle[1].

**Scope.** This work focuses exclusively on *model-level provenance inference*, which we identify as a core and tractable component of the broader AI supply chain. CHAININFER aims to recover missing or latent relationships among models, including fine-tuning, merging, quantization, and other transformation-induced dependencies, since these relationships form the backbone of provenance flows in modern model ecosystems. Other AI supply chain elements, including datasets, software libraries, training pipelines, and hardware environments, are outside the current scope due to the lack of standardized metadata. Nonetheless, CHAININFER establishes a foundation for scalable, graph-based inference methodologies that can be extended in future work to incorporate non-model entities, enabling a more comprehensive and end-to-end AI supply chain analysis.

## 2 BACKGROUND AND RELATED WORKS

**AI supply chain provenance.** Recent efforts have focused on collecting and organizing provenance information in AI supply chains. Tools and studies leverage platform APIs, metadata files, model

---

[1]Both graphs will be shared publicly upon acceptance.

cards, and repository scans to construct partial lineage graphs (Mitchell et al., 2023; Xu et al., 2023; Rahman et al., 2025), identifying relations such as fine-tuning, quantization, adaptation, and merging. While these works provide valuable foundations, they largely emphasize data acquisition and visualization, leaving open the problem of predictive inference for missing or latent relationships.

**Model lineage reconstruction** has been studied through direct comparison of parameters and weight files (Birhane et al., 2023; Shen et al., 2021), analysis of fine-tuning adapters or LoRA modules (Hu et al., 2022; Pfeiffer et al., 2020), and mining of training logs or dataset references (Geiger et al., 2020; Jo & Gebru, 2020). These methods can recover explicit provenance edges when metadata is available but rely on deterministic heuristics or static parsing, which limits their ability to generalize. By contrast, our work formulates AI supply chain analysis as a predictive graph problem, aiming to infer both connectivity and relationship directly from graph structure and attributes.

**Graph neural networks (GNNs)** learn representations from graph-structured data by iteratively aggregating and transforming information from a node's neighborhood (Scarselli et al., 2008). Through repeated message-passing layers, a node embedding encodes both its own attributes and contextual information from its surrounding structure. Classical variants differ in how they aggregate messages: Graph Convolutional Networks (GCN) (Kipf & Welling, 2016) employ spectral filtering, Graph Attention Networks (GAT) (Veličković et al., 2017) leverage attention mechanisms, and Graph Isomorphism Networks (GIN) (Xu et al., 2018) are designed to match the expressive power of the Weisfeiler–Lehman test. GNNs have achieved strong performance on node classification, link prediction, and graph-level classification.

**Graph transformers (GTs).** While GNNs propagate information locally, their reliance on neighborhood aggregation can lead to issues such as over-smoothing and over-squashing. GTs extend transformer architectures to graphs by allowing each node to attend globally to all others. GT (Dwivedi & Bresson, 2020) is the first introduced Graph Transformer, incorporating structural encodings such as Laplacian eigenvectors to preserve positional information and allow the model to reason about global structure. Building on this, Graphormer (Ying et al., 2021a) uses shortest path distance encoding and centrality-based attention bias to more effectively capture both connectivity patterns and global relationships. Note that, GTs remain more computation intensive than message-passing GNNs as global attention introduces quadratic complexity in the number of nodes.

**Missing metadata inference in other domains.** Although the AI supply chain is an emerging area, the task of inferring missing provenance has clear parallels in software engineering and knowledge graph research. In software ecosystems such as npm, PyPI, and Software Heritage, software artifacts are modeled as dependency graphs, and prior work has investigated recovering missing or undocumented dependencies through static analysis, collaborative filtering, or graph-based representation learning Zhu & Zimmermann (2018); Mirhosseini & Parnin (2017); Abate et al. (2020). Related efforts in software supply chain security have framed dependency prediction and vulnerability propagation as link prediction problems on large heterogeneous graphs Zimmermann et al. (2019); Pashchenko et al. (2020). More broadly, provenance inference in interconnected systems has been studied within the knowledge graph completion literature, where models such as GCN, GAT, and GIN are widely used for relational link prediction and structural inference Kipf & Welling (2017); Velickovic et al. (2018); Xu et al. (2019b). These research directions collectively demonstrate that inferring missing connections in complex artifact graphs is commonly approached through graph completion or link prediction techniques across several domains.

## 3 METHODOLOGY: CHAININFER

### 3.1 PROBLEM STATEMENT

We model the *AI supply chain* as an *attributed, typed, directed* graph $G = (V, E, \mathcal{T}_V, \mathcal{T}_E, X)$, where $V$ is the set of nodes ($N = |V|$), each node $v \in V$ represents an AI model, and each node carries an attribute vector $x_v \in \mathbb{R}^d$ obtained by concatenating metadata features that reflect complementary aspects of supply chain provenance. The set of directed edges $E \subseteq V \times V$ ($M = |E|$) encodes provenance relations or supply chain dependencies. The edge type mapping $\mathcal{T}_E : E \to \mathcal{C}_E$ assigns each edge to a semantic relation (e.g., `fine-tuned`, `adapted`, `quantized`, `merged`).

Table 1: Extracted features from AI models towards AI supply chain semantics.

| Category | Extracted features |
|---|---|
| Model architecture | depth (#layers), hidden size, #attention heads, vocab size, positional-encoding type, encoder/decoder family |
| Tokenizer & preprocessing | tokenizer type (BPE/SentencePiece/WordPiece), vocab files and sizes, special/added tokens, normalization rules |
| Quantization & pruning | quantized flag, bits-per-weight (weights/activations), scheme (e.g., GPTQ/AWQ), weight format (int4/int8), sparsity ratio/pattern |
| File-level metadata | framework/export (PyTorch/ONNX/gguf/safetensors), version, file counts/sizes (log-scaled), license tag, card/README embedding |
| Model fingerprints | layer-wise weight histograms, random projections/sketches, checksum families, checkpoint lineage IDs |
| PEFT | adapter presence, type (LoRA/IA3/prefix), ranks/alphas/target modules |

Given a *partially observed AI supply chain graph* $G_{\text{obs}} = (V_{\text{obs}}, E_{\text{obs}})$ and a *new model* $u \notin V_{\text{obs}}$, the problem of *AI supply chain information inference* consists of the following two tasks.

**Problem #1: Link prediction:** Predict which existing models in $V_{\text{obs}}$ are potential dependencies of the new model $u$. That is, predict a set of candidate edges $\hat{E}_u = \{(u, v) \mid v \in V_{\text{obs}}, (u, v) \notin E_{\text{obs}}\}$, where each predicted edge $(u, v)$ represents a plausible provenance relation.

**Problem #2: Edge classification:** Determine the semantic type of each predicted edge. This is formalized as a function $\mathcal{F}_E : \hat{E}_u \to \mathcal{C}_E$, which assigns a dependency label (fine-tuned, quantized, adapted, merged) to each edge.

### 3.2 FEATURE EXTRACTION TOWARDS AI SUPPLY CHAIN SEMANTICS

**Extracted features.** We extract the features from models that can capture the complementary signals related to the AI supply chain. The details of how to extract features can be found in Appendix C. The features are classified into six feature blocks as shown in Table 1. (i) *Model architecture* captures quasi-invariant structural design choices such as depth, hidden size, and attention configuration. These properties are typically preserved under fine-tuning and parameter-efficient adaptation, making them strong ancestry cues for link prediction. Conversely, significant divergences in architecture (e.g., altered layer counts or cross-family hybrids) signal merges or re-architecting events, thereby informing edge classification.

(ii) *Tokenizer & preprocessing* record vocabulary artifacts, tokenizer families, and normalization strategies that constrain model compatibility. Shared vocabularies often indicate overlapping datasets or domains, while the presence of added or remapped tokens marks downstream adaptation. These signals support link prediction by clustering models with common data foundations and guide edge classification by exposing adaptation-specific footprints. (iii) *Quantization and pruning* provide distinctive *transform footprints* of post-training compression (e.g., bits-per-weight, schemes, and sparsity), directly supporting quantized edges. (iv) *File-level metadata* (e.g., framework/export, versions, licenses, and README) traces toolchain lineage and frequently references source checkpoints/datasets, helping bridge documentation gaps. (v) *Model fingerprints* (e.g., histograms, sketches, and checksums) enable fine-grained reuse detection (e.g., capturing minor edits, merges, and derivations) and thus improve parent discovery. (vi) *Parameter-efficient fine-tuning (PEFT) indicators* (e.g., adapter types/configurations, and merge state) are explicit evidences for adapted/fine-tuned relations, distinguishing them from merged/quantized.

**Per-block encoding and normalization.** Each block $b \in \{1, \ldots, 6\}$ is mapped to a fixed-width vector $\phi_b \in \mathbb{R}^{d_b}$, and we set $x_v = [\phi_1 \parallel \phi_2 \parallel \phi_3 \parallel \phi_4 \parallel \phi_5 \parallel \phi_6] \in \mathbb{R}^d$, where $d = \sum_b d_b$. *Numeric* fields (e.g., hidden size, file size, and sparsity) are log- or z-normalized; *categorical* fields (e.g., framework, tokenizer type) use learned embeddings; *textual* fields (README/card) use a frozen text encoder sentence-transformer (Reimers & Gurevych, 2019) to produce a dense vector; *fingerprints* use robust sketches (checksum families and low-rank projections aggregated across layers) to stabilize similarity under minor edits or quantization. When entries are missing, we use the missingness-aware pipeline described later (Section 3.4) so the encoder can treat observed vs. imputed values differently.

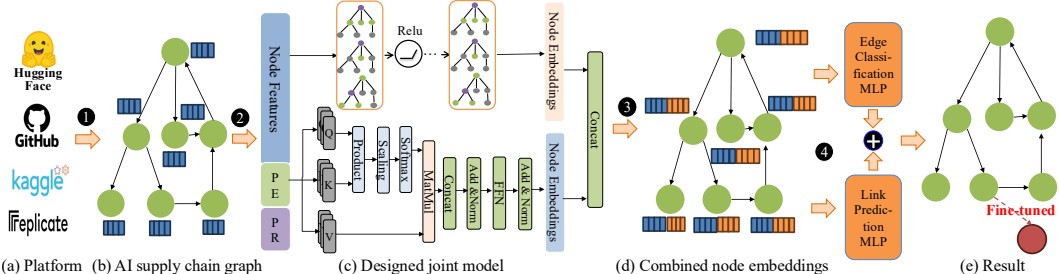

(a) Platform    (b) AI supply chain graph      (c) Designed joint model      (d) Combined node embeddings      (e) Result

Figure 2: Overview of CHAININFER. From the AI platforms, ❶ we build the AI supply chain graph, ❷ we apply the joint modeling method and get the node/edge embeddings ❸, and ❹ we jointly apply the node/edge embeddings to the tasks of link prediction and edge classification.

### 3.3 JOINT MODELING FOR LINK PREDICTION AND EDGE CLASSIFICATION

Our goal is to recover missing links between models (*link prediction*) and to assign semantic labels to observed dependencies (*edge classification*). Both tasks are challenging as they require reasoning across *multiple scales*. That is, local neighborhoods encode fine-grained cues such as quantization or tokenizer metadata, while long-range provenance chains capture global context, e.g., whether two models ultimately derive from the same foundation checkpoint. Relying solely on local message passing risks missing these global dependencies due to over-squashing, whereas using only global attention is computationally prohibitive and overlooks fine local distinctions. This motivates our hybrid design that integrates GNN layers for efficient local encoding with GT layers for global reasoning. Figure 2 provides an overview of the proposed CHAININFER.

**Input projection and positional structure.** In particular, the graph model needs to simultaneously capture *node-level attributes* and *structural context*. Let $X = [x_1, \ldots, x_N] \in \mathbb{R}^{N \times d}$ be node features and $P \in \mathbb{R}^{N \times k}$ the top-$k$ eigenvectors of the normalized Laplacian $L_{\text{sym}} = I - D^{-1/2} A D^{-1/2}$, which serve as structural positional encodings. We first map features into a hidden space and inject structural bias:

$$H^{(0)} = \text{ReLU}(XW^{(0)} + b^{(0)}), \qquad H^{(0)} \leftarrow H^{(0)} + MLP(P), \tag{1}$$

with $W^{(0)} \in \mathbb{R}^{d \times d_h}$. This projection ensures that the encoder has access to both semantic attributes and global geometry from the outset.

**Local GNN layers.** While global attention is vital for capturing long-range provenance, many of the most informative signals in the AI supply chain graph are inherently *local*. For link prediction, strong cues for the existence of an edge often come from neighborhood-level structure. For example, if a model and its suspected parent share tokenizer settings, quantization metadata, or fine-tuning adapters, their local neighborhoods strongly suggest a direct dependency. For edge classification, distinguishing relation types such as `quantized` versus `fine-tuned` frequently hinges on subtle feature differences in the immediate vicinity of nodes, such as architecture fingerprints or PEFT.

Graph neural network (GNN) layers, particularly Graph Isomorphism Networks (GIN) Xu et al. (2018), are well-suited for encoding these high-frequency, neighborhood-level patterns because of their strong discriminative power over local multisets. By aggregating and transforming features within a few hops, GNN layers provide expressive local embeddings that capture the fine-grained metadata differences crucial for both tasks, while also serving as an efficient preconditioner that reduces the burden on global GT layers.

In particular, we use GIN due to its strong expressive power among GNNs. GIN reliably distinguishes different neighborhood structures (provably matching the 1-WL test, detailed analysis in Appendix B), which is crucial for discriminating subtle local provenance patterns in the AI supply chain graphs. We stack $L_{\text{GIN}}$ layers:

$$h_i^{(\ell+1)} = \text{MLP}^{(\ell)}\Big((1 + \varepsilon^{(\ell)}) h_i^{(\ell)} + \sum_{j \in \mathcal{N}(i)} h_j^{(\ell)}\Big), \quad \ell = 0, \ldots, L_{\text{gin}} - 1, \tag{2}$$

with learnable $\varepsilon^{(\ell)}$ and residual connections and normalization applied. We denote the final GIN outputs by $H_{\mathrm{GIN}}$.

**Global GT layers.** In the AI supply chain graph, many dependencies are inherently long-range. For instance, whether a link exists between two models (*link prediction*) may depend on their shared ancestry several hops away, such as originating from the same base checkpoint or dataset. Similarly, determining the *type* of a relation (*edge classification*) often requires reconciling signals across multiple, distant steps of provenance. For example, distinguishing a `fine-tuned` from a `merged` may hinge on observing parallel adaptation paths or cross-model integration patterns. Purely local message-passing GNN layers struggle in this regime. That is, increasing depth leads to over-squashing, where exponentially many distant messages are compressed into small vectors, and over-smoothing, where node embeddings become indistinguishable.

GT layers with global attention address this by creating direct, content-aware communication channels between far-apart nodes, guided by structural positional encodings. Therefore, we add a few GT layers in our joint modeling, which can capture these global provenance cues that are essential for accurate link prediction and reliable edge classification. Particularly, GT layers use global multi-head self-attention.

$$\tilde{H}^{(\ell)} = \mathrm{MHA}\big(H^{(\ell)}\big) + H^{(\ell)}, \qquad H^{(\ell+1)} = \mathrm{FFN}\big(\tilde{H}^{(\ell)}\big) + \tilde{H}^{(\ell)}, \tag{3}$$

for $\ell = 0, \ldots, L_{\mathrm{gt}} - 1$, with final outputs $H_{\mathrm{GT}}$. These layers allow direct interaction between distant nodes, mitigating information loss along multi-hop paths. However, the cost per GT layer is

$$\mathrm{Time} = \mathcal{O}(N^2 d_h), \qquad \mathrm{Memory} = \mathcal{O}(N^2), \tag{4}$$

which motivates using only $1-2$ GT layers and optionally restricting their hidden width $d_h^{\mathrm{GT}} \le d_h$. This retains most of the accuracy benefits of global reasoning while controlling quadratic overhead.

**Hybrid fusion and prediction.** Together, we combine local encoding $h_i$ and global encoding $h_j$.

$$\hat{y}_{(i,j)}^{\mathrm{label}} = \mathrm{Softmax}(\mathrm{MLP}_{\mathrm{label}}([h_i \| h_j])), \qquad \hat{y}_{(i,j)}^{\mathrm{link}} = \sigma(\mathrm{MLP}_{\mathrm{link}}([h_i \| h_j])). \tag{5}$$

The training objective jointly optimizes both tasks:

$$\mathcal{L} = \lambda\, \mathcal{L}_{\mathrm{link}} + (1 - \lambda)\, \mathcal{L}_{\mathrm{label}}, \lambda \in [0, 1] \tag{6}$$

where $\mathcal{L}_{\mathrm{link}}$ is binary cross-entropy for link prediction and $\mathcal{L}_{\mathrm{label}}$ is cross-entropy for edge classification. Sharing the encoder across tasks encourages it to preserve information useful for both link existence and relation typing.

**Complexity analysis.** Let $C_{\mathrm{GT}}(L, d, N) = L \cdot c_1 N^2 d$ denote the dominant cost of a GT stack. A GT-only encoder achieving target accuracy typically requires $(L^*, d^*)$, which is computationally heavy. Our hybrid achieves similar accuracy with only $(L_{\mathrm{gt}}, d_{\mathrm{GT}})$, where $L_{\mathrm{gt}} \ll L^*$ and $d_{\mathrm{GT}} \le d^*$, since GIN already resolves local dependencies. The overall complexity is

$$\mathrm{Time} \approx L_{\mathrm{GIN}}\, \mathcal{O}(M d_h)\ +\ L_{\mathrm{GT}}\, \mathcal{O}(N^2 d_h)\ +\ \mathcal{O}(N d_h^2), \tag{7}$$

Where $O(N d_h^2)$ is the readout/MLP layer cost. This is substantially smaller than a GT-only model, as the GIN branch offloads local computation, allowing GT layers to focus narrowly on global interactions at modest depth and width.

## 3.4 Cold-Start Robustness and Missing Metadata

Public AI supply chain corpora are often incomplete, i.e., many models arrive with partially filled or entirely missing metadata fields. This creates *cold-start* nodes whose features are unreliable or absent, yet our tasks (*link prediction* and *edge classification*) still require accurate reasoning about their provenance. A robust encoder must therefore (i) exploit whatever attributes are present, (ii) leverage structural signals from the graph topology, and (iii) propagate evidence from informative neighbors and distant ancestors.

**Missingness-aware encoding and imputation.** Let $x_v \in \mathbb{R}^d$ be the raw features of node $v$, and $m_v \in \{0, 1\}^d$ a binary mask (1 = observed). We learn an imputer $g_\theta$ that conditions on observed entries and structural context (e.g., degrees, and positional encodings):

$$\hat{x}_v = g_\theta\big(x_v \odot m_v\,;\, \mathrm{struct}(v)\big), \qquad \tilde{x}_v = (x_v \odot m_v) + (1 - m_v) \odot \hat{x}_v, \qquad \tilde{x}_v \leftarrow [\,\tilde{x}_v \,\|\, m_v\,]. \tag{8}$$

Table 2: Performance and time (seconds per epoch). The best results are **bolded**.

| Category | Model | $\text{Acc}_{\text{edge}}\uparrow$ | $\text{F1}_{\text{edge}}\uparrow$ | $\text{AUC}_{\text{edge}}\uparrow$ | $\text{Acc}_{\text{link}}\uparrow$ | $\text{F1}_{\text{link}}\uparrow$ | $\text{AUC}_{\text{link}}\uparrow$ | $\text{Acc}_{\text{joint}}\uparrow$ | $\text{Time}\downarrow$ |
|---|---|---|---|---|---|---|---|---|---|
| GNN-only | GCN | 0.60±0.04 | 0.62±0.03 | 0.65±0.05 | 0.68±0.03 | 0.67±0.02 | 0.71±0.03 | 0.72±0.03 | 0.08±0.02 |
| | GAT | 0.42±0.02 | 0.41±0.04 | 0.59±0.03 | 0.63±0.04 | 0.62±0.02 | 0.66±0.02 | 0.70±0.03 | 0.19±0.07 |
| | GIN | 0.66±0.02 | 0.68±0.03 | 0.71±0.03 | 0.73±0.02 | 0.73±0.03 | 0.76±0.02 | 0.74±0.04 | **0.06**±0.02 |
| GT-only | Graphormer | 0.78±0.02 | 0.83±0.03 | 0.84±0.03 | 0.86±0.04 | 0.84±0.02 | 0.80±0.02 | 0.86±0.03 | 2.19±0.12 |
| | GT | 0.76±0.02 | 0.81±0.03 | 0.83±0.03 | 0.84±0.03 | 0.84±0.03 | 0.79±0.03 | 0.84±0.03 | 1.80±0.14 |
| Separate | GCN+Graphormer | 0.79±0.04 | 0.79±0.02 | 0.85±0.04 | 0.83±0.03 | 0.86±0.02 | 0.87±0.03 | 0.84±0.03 | 2.41±0.09 |
| | GAT+Graphormer | 0.80±0.02 | 0.80±0.03 | 0.83±0.04 | 0.84±0.04 | 0.85±0.03 | 0.88±0.03 | 0.86±0.02 | 2.45±0.08 |
| | GIN+Graphormer | 0.81±0.04 | 0.80±0.02 | 0.85±0.03 | 0.85±0.02 | 0.86±0.03 | 0.87±0.03 | 0.89±0.03 | 2.34±0.15 |
| | GCN+GT | 0.78±0.03 | 0.79±0.03 | 0.82±0.03 | 0.84±0.02 | 0.87±0.03 | 0.86±0.03 | 0.84±0.04 | 1.82±0.07 |
| | GAT+GT | 0.77±0.01 | 0.82±0.03 | 0.83±0.03 | 0.83±0.02 | 0.83±0.04 | 0.86±0.03 | 0.87±0.03 | 2.01±0.09 |
| | GIN+GT | 0.77±0.03 | 0.82±0.04 | 0.84±0.03 | 0.85±0.02 | 0.86±0.03 | 0.85±0.03 | 0.86±0.02 | 1.85±0.07 |
| Joint | GCN+Graphormer | 0.89±0.03 | 0.91±0.05 | 0.92±0.02 | 0.91±0.02 | 0.92±0.01 | 0.92±0.02 | 0.91±0.02 | 3.77±0.06 |
| | GAT+Graphormer | 0.88±0.04 | 0.91±0.04 | 0.93±0.02 | 0.91±0.03 | 0.92±0.04 | 0.93±0.02 | 0.90±0.04 | 4.14±0.10 |
| | GIN+Graphormer | 0.90±0.04 | 0.91±0.04 | 0.92±0.03 | **0.92**±0.04 | **0.94**±0.03 | 0.93±0.01 | 0.91±0.02 | 3.51±0.09 |
| | GCN+GT | 0.87±0.02 | 0.92±0.03 | 0.94±0.02 | 0.91±0.02 | 0.91±0.01 | 0.91±0.02 | 0.91±0.03 | 2.57±0.11 |
| | GAT+GT | 0.88±0.02 | 0.92±0.02 | 0.94±0.03 | 0.90±0.02 | 0.93±0.03 | **0.95**±0.01 | 0.90±0.02 | 2.80±0.11 |
| | CHAININFER | **0.94**±0.02 | **0.93**±0.01 | **0.96**±0.02 | 0.91±0.03 | **0.94**±0.02 | 0.94±0.01 | **0.94**±0.03 | 2.34±0.06 |

Concatenating $m_v$ exposes missingness as a first-class signal so the encoder can *treat imputed values differently from observed ones*. We train $g_\theta$ with a denoising-style objective by randomly masking observed entries and reconstructing them (for categorical fields, cross-entropy over codes; for continuous, scaled $\ell_1/\ell_2$), and combine this auxiliary loss with the joint task loss.

**Topology-derived priors.** Even when attributes are scarce, the graph carries informative priors. We append normalized in/out/total degree and PageRank $\pi_v$ to the input (together with Laplacian positional encodings already injected in the hybrid encoder). High-degree or high-$\pi_v$ nodes often correspond to widely reused base checkpoints or datasets, while low-degree nodes tend to be leaves.

**Joint modeling for cold start.** GIN layers aggregate neighborhood evidence, allowing well-described neighbors to "lend" information to poorly described nodes and to preserve fine-grained local patterns needed to disambiguate relation types (e.g., fine-tune vs. quantization). Global GT layers then create content-aware shortcuts across multi-hop provenance, which is crucial when a cold-start node's immediate neighbors are also sparse: attention can pull in signals from distant but relevant ancestors or siblings. Thus, the same local-global complementarity that benefits fully observed nodes is *especially* valuable under missingness.

## 4 EXPERIMENTS

### 4.1 EXPERIMENTAL SETUP

We implement CHAININFER using PyTorch Geometric (PyG) (Fey & Lenssen, 2019) and train all models on our in-house server. The server is equipped with four NVIDIA L40S GPUs, each with 40 GB of memory, although only a single GPU is used for each experiment. The system contains two AMD EPYC 9334 processors, each with 32 cores and hyperthreading enabled, for a total of 128 logical threads. All models are implemented in PyTorch 2.0.1 with CUDA 11.8, and we use PyG 2.6.1 for graph operations. Unless otherwise noted, all baselines are implemented within the same framework and are trained under identical hardware and optimizer settings to ensure fair comparison.

**AI supply chain graph construction.** We construct the AI supply chain graph from the Hugging Face platform (Hugging Face) with four steps, including model tree extraction, model card analysis, structured metadata, and repository scan and checkpoints. More details can be found in Appendix C. This yields a directed attributed graph with **200,000** nodes and **193,016** edges. Each node represents a model, and edges are labeled with one of four provenance types: fine-tuned, quantized, adapted, or merged. Particularly, 41.5% are adapted, 34.2% are fine-tuned, 0.3% are merged, and 24.1% are quantized.

**Training protocol.** We adopt a transductive setting: all nodes are visible, while edges are split at the relation level into 70% train, 10% validation, and 20% test (stratified by relation to preserve class balance). For link prediction, we sample an equal number of non-edges as negatives within

each split to balance positives/negatives. Models are trained with Adam (learning rate $1 \times 10^{-3}$, weight decay $1 \times 10^{-5}$) and early stopping on validation loss with a patience of 20 epochs. Unless specified, *GNN baselines* use two message-passing layers with hidden size 128 and dropout 0.3, and *GT baselines* use two encoder layers with four attention heads and Laplacian positional encodings (top $k = 8$ eigenvectors of the normalized Laplacian), also with dropout 0.3. For CHAININFER, we fuse local and global information by concatenating the embeddings from a two-layer GIN branch and a two-layer GT branch, followed by a shared MLP head. A single set of node embeddings supports both tasks, i.e., link prediction (edge existence) and edge classification (relation type).

**Evaluation metrics.** We report Accuracy (Acc), micro F1-score, and ROC-AUC (AUC). As supply chain reconstruction is inherently *end-to-end*, a system is only correct when it both finds the right neighbor and assigns the right relation. In addition, we report a *joint accuracy* that couples link prediction and edge classification. Let $\hat{\mathcal{E}} = \{(u,v) \mid s_{uv} \geq \tau\}$ be the predicted edge set at test time, $\mathcal{E}$ the ground-truth edges, and $y_{(u,v)}$ the ground-truth relation. We define $\text{Acc}_{\text{joint}} = \frac{\sum_{(u,v) \in \hat{\mathcal{E}}} \mathbf{1}((u,v) \in \mathcal{E}) \mathbf{1}(\hat{y}_{(u,v)} = y_{(u,v)})}{|\hat{\mathcal{E}}|}$. This metric counts an edge as correct *iff* the model (i) predicts its existence and (ii) assigns the correct relation, thereby aligning the metric with the practical objective of provenance reconstruction. Differently, the joint accuracy takes the results from both link prediction and edge classification into consideration. It is an end-to-end metric that uses the test set only once and counts an edge as correct only if the model both predicts that the edge exists and assigns the correct class. Thus, joint accuracy is strictly more demanding than either individual metric, as any error in link prediction or type prediction results in failure. This metric best reflects real-world usage, where a system must detect a relationship and correctly classify its type simultaneously.

## 4.2 PERFORMANCE COMPARISON

**Compared methods.** We benchmark CHAININFER against strong and representative graph learning families that instantiate complementary inductive biases (*local message passing* vs. *global attention*) and an explicit *non-joint hybrid* to test whether simple combination already suffices. All of them use the same node/edge features, data splits, and training protocol. The depth, hidden size, and head counts are tuned under a matched parameter budget (within 5%) with early stopping on validation performance. This setup ensures a fair, apples-to-apples comparison against widely accepted, high-quality baselines. In particular, we test different models in four categories as discussed below.

(i) *GNN-only*: We include GCN (Kipf & Welling, 2016), GAT (Veličković et al., 2017), and GIN (Xu et al., 2018). These message-passing models are widely reported as competitive on large graphs (e.g., OGB benchmarks (Hu et al., 2020)) for both edge classification and link prediction. (ii) *GT-only*: To assess the benefit of global context, we evaluate a global GT with Laplacian eigenvector positional encodings (Dwivedi & Bresson, 2020), and another variant Graphormer (Ying et al., 2021a) of using shortest paths for positional encoding. (iii) *Separately trained models*: To understand the effect of joint learning compared with simple model combination, we trained the GNN and GT variants separately on the same dataset, ensuring no parameter sharing or joint optimization. After independent training, we combine their outputs in late fusion, where we concatenate the learned embeddings from both models and passed them through an MLP for prediction. To this end, we get six separately trained models. (iv) *Jointly trained models*: As CHAININFER is a jointly trained model with GIN and GT layers, to understand the effect of other combinations, we also test additional five jointly trained models of GNN and GT variants. They share the same setting with CHAININFER.

Table D summarizes the results. We made three observations. (i) *Jointly trained hybrids deliver the strongest overall results*. By co-optimizing GNN and GT branches, models achieve large performance gains, with edge and link metrics exceeding 0.90 across the board. Our CHAININFER attains the highest scores on all metrics, reaching 0.93 joint accuracy, while also reducing training time relative to other joint-training variants. This demonstrates that integrating local message passing with global attention in a unified architecture is critical for accurate AI supply chain inference.

(ii) *Separately trained GNN+GT models* improve over either family alone, indicating that local and global features are complementary. Nonetheless, their joint accuracies plateau around 0.87-0.89, suggesting that naive fusion (e.g., late concatenation or logit averaging) does not fully exploit the interaction between local and global contexts.

(iii) *GNN-only baselines* (GCN, GAT, GIN) achieve competitive performance on local neighborhood modeling but struggle to capture multi-hop and global dependencies inherent in AI supply chains. Among them, GIN performs best, consistent with its higher expressiveness relative to GCN/GAT, yet its joint accuracy remains below $0.76$. *GT-only models* (GT, Graphormer) substantially outperform GNNs across all metrics benefited from global attention and positional encodings to capture long-range dependencies, while their training times are much slower.

### 4.3 ABLATION STUDY

We perform two sets of ablations to better understand the design choices behind CHAININFER: (i) architectural parameters, and (ii) the impact of structural features. We also study the impact of varying AI supply chain graph sizes, which can be found in Appendix D.

**Architectural parameters.** Table 3 summaizes the results of varying GIN layers ($L$), Transformer depth ($D$), and attention heads ($h$), while fixing the others to their defaults ($L=2$, $D=2$, $h=4$). We observe that, (i) increasing GIN layers beyond two provides no additional joint-accuracy gains. That is, moving from $L=2$ to $L=3$ raises link accuracy slightly ($0.93$ to $0.94$) but leaves joint accuracy unchanged at $0.93$, while increasing runtime from $2.33$ s/epoch to $2.39$ s/epoch. (ii) Similarly, deeper Transformers improve performance up to $D=2$ but become memory-

Table 3: Architectural ablations (best are **bolded**).

| Variant | $Acc_{edge}\uparrow$ | $Acc_{link}\uparrow$ | $Acc_{joint}\uparrow$ | $Time\downarrow$ |
|---|---|---|---|---|
| GIN layers $L=1$ | 0.91 | 0.91 | 0.93 | 2.22 |
| **GIN layers $L=2$** | **0.92** | **0.92** | **0.93** | **2.33** |
| GIN layers $L=3$ | 0.92 | 0.94 | 0.93 | 2.39 |
| GT depth $D=1$ | 0.90 | 0.91 | 0.90 | 2.07 |
| **GT depth $D=2$** | **0.92** | **0.92** | **0.93** | **2.33** |
| GT depth $D=3$ | OOM | OOM | OOM | OOM |
| GT heads $h=1$ | 0.90 | 0.91 | 0.91 | 2.02 |
| **GT heads $h=4$** | **0.92** | **0.92** | **0.93** | **2.33** |
| GT heads $h=6$ | 0.92 | 0.94 | 0.93 | 3.14 |

prohibitive at $D=3$, highlighting the quadratic cost of global attention. (iii) Multi-head attention shows diminishing returns. That is, $h=4$ attains the best balance between accuracy and efficiency ($0.93$ joint accuracy at $2.33$ s/epoch), while $h=6$ increases runtime to $3.14$ s/epoch without further joint-accuracy improvement. These results indicate that moderate depth and head count suffice to capture both local and global context, while excessive capacity incurs steep computational overhead.

**Structural features.** Table 4 summarizes the effect of structural features. A baseline CHAININFER without structural features (base) achieves $0.88$ joint accuracy, which remains unchanged when augmented with random-walk Personalized PageRank (PR) features. Incorporating Laplacian positional encodings (LapPE), however, raises joint accuracy to $0.91$ by injecting richer spectral information about

Table 4: Effect of structural features (best are **bolded**).

| Variant | $Acc_{edge}\uparrow$ | $Acc_{link}\uparrow$ | $Acc_{joint}\uparrow$ |
|---|---|---|---|
| Base | 0.89 | 0.89 | 0.88 |
| Base+PR | 0.90 | 0.90 | 0.88 |
| Base+LapPE | 0.91 | 0.92 | 0.91 |
| **Base+PR+LapPE** | **0.92** | **0.92** | **0.93** |

global graph structure. CHAININFER, which integrates LapPE with additional handcrafted structural descriptors (e.g., degree, and centrality statistics), further improves to $0.93$ joint accuracy, confirming that combining learned embeddings with principled structural priors significantly enhances provenance inference.

### 4.4 INDUCTIVE GENERALIZATION TO OTHER AI PLATFORMS

To evaluate the inductive capability, we train CHAININFER on the Hugging Face graph and directly apply it to the Kaggle graph, which is completely disjoint and unseen during training. Both graphs share the same node/edge feature schema and relation taxonomy, but differ in ecosystem conventions such as metadata quality, naming patterns, and repository organization. For Kaggle evaluation, we freeze model parameters, recompute structural encodings locally, and use the same validation threshold for link prediction, ensuring a strict zero-shot transfer with no target-domain tuning.

Table 5 summarizes the results. When applied inductively to Kaggle, CHAININFER retains strong performance with **0.90 joint accuracy**, a modest 3-point drop compared to transductive inference on Hugging Face. Link accuracy remains stable ($0.92 \rightarrow 0.91$), while edge-type accuracy decreases

Table 5: Cross-platform inference time (seconds) and accuracy (best are **bolded**).

| Graph | $Acc_{edge}\uparrow$ | $Acc_{link}\uparrow$ | $Acc_{joint}\uparrow$ | $Time\downarrow$ |
|---|---|---|---|---|
| Hugging Face | 0.92 | 0.92 | 0.93 | 0.41 |
| **Kaggle** | **0.89** | **0.91** | **0.90** | **0.23** |

slightly more (0.92 → 0.89), indicating that relation label is more sensitive to platform-specific conventions. These results confirm that CHAININFER generalizes effectively across AI platforms.

## 4.5 USE CASE

To understand how CHAININFER helps in practice, let us study an example, i.e., the release of a new model *microsoft/VibeVoice-1.5B* (Microsoft Inc., 2025) from Microsoft, where the metadata is incomplete or ambiguous. In the original repository, the model card may omit its training lineage or downstream adaptations, leaving risk assessors and auditors with limited visibility. With CHAININFER, the new model is first added as an isolated node without edges. CHAININFER then generates node embeddings by combining local structural cues (via GIN layers) with global relational context (via GT layers). Link prediction scores are computed between this node and all existing nodes in the supply chain graph. Then, the high scoring candidates are flagged as plausible dependencies. In this example, CHAININFER recovers two edges: one pointing to *omarViga/speecht5-tts-mabam-es* and another to *wsbagnsv1/VibeVoice-1.5B-gguf*. The edge-classification module then assigns semantic labels, identifying the first as a `fine-tuned` relation and the second as a `quantized` relation, precisely matching the ground-truth annotations later confirmed by the developers.

This pipeline highlights the practical value of CHAININFER: it reconstructs both connectivity and relation types for models whose provenance is only partially specified. More broadly, this method can scale to thousands of new releases per month across platforms such as Hugging Face and Kaggle. By continuously updating the AI supply chain graph, CHAININFER enables downstream applications, such as automated vulnerability tracing, compliance auditing, and ecosystem monitoring. CHAININFER provides a foundation for reliable, transparent, and scalable AI supply chain intelligence.

## 5 CONCLUSION

We presented CHAININFER, a hybrid GNN and GT model for inferring missing structure in AI supply chains. By coupling link prediction with edge classification on large attributed graphs, CHAININFER achieves state-of-the-art accuracy while remaining efficient and transferable. Experiments on Hugging Face and Kaggle demonstrate its scalability and robustness, positioning CHAININFER as a practical tool for automated provenance reconstruction in real-world AI platforms.

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

## A  USAGE OF LLM

During the preparation of this manuscript, we used the GPT-5 mini model solely for grammar correction and language editing. Specifically, prompts such as "Are there any grammatical errors in this paragraph?" and "Which parts of this paragraph can be removed?" were employed. The LLM model was not used for any other purposes.

## B  1–WL EXPRESSIVITY ANALYSIS AND MODEL DESIGN

The 1–Weisfeiler–Lehman (1–WL) color refinement test (Weisfeiler & Lehman, 1968) offers a principled lens for reasoning about the expressive power of message-passing GNN layers. Starting from initial node labels (or features), 1–WL iteratively updates each node color $c_i^{(\ell)}$ by hashing the multiset formed by its current color and the colors of its neighbors.

$$c_i^{(\ell+1)} \leftarrow \text{Hash}\Big(c_i^{(\ell)}, \ \{\{c_j^{(\ell)} : j \in \mathcal{N}(i)\}\}\Big) \tag{9}$$

It declares two graphs non-isomorphic if their multisets of colors differ at any iteration. Message-passing GNNs are upper-bounded by 1–WL in distinguishing non-isomorphic graphs; they *match* 1–WL when each layer computes an *injective* permutation-invariant function of the multiset of neighbor features together with the node's own feature (Xu et al., 2019a; Barceló et al., 2021). By the standard representation of invariant set functions (Zaheer et al., 2017), any injective multiset map can be written (up to capacity) as the following equation for suitable transformations $\phi$ and $\rho$.

$$F\big(\{\mathbf{x}_1, \dots, \mathbf{x}_k\}\big) \ = \ \rho\left(\sum_{i=1}^{k} \phi(\mathbf{x}_i)\right) \tag{10}$$

This motivates sum-aggregation with an MLP, as in GIN (Xu et al., 2019a), to attain 1–WL-level discrimination on local neighborhoods.

In AI supply chain graphs, nodes carry heterogeneous metadata (architecture, tokenizer, quantization/pruning, PEFT indicators), and edges encode provenance relations (e.g., fine-tuned, merged, adapted, quantized). Accurate *link prediction* requires detecting whether local neighborhoods are consistent with a direct derivation, while *edge classification* must discriminate relation types that can differ only by subtle local cues. Non-injective aggregators (e.g., mean pooling or normalized attention with fixed stencil) can collapse distinct multisets to identical representations, obscuring exactly these local differences and harming both tasks. GIN's injective multiset aggregation, coupled with non-linear updates, preserves fine-grained distinctions among neighborhoods, making it well-suited to capture local provenance motifs such as short fine-tuning chains or adapter placements.

However, 1–WL (and thus any purely local message-passing stack) is inherently limited in capturing *long-range* dependencies: distinguishing patterns that require reconciling evidence across multiple hops can demand many iterations, exacerbating over-squashing and over-smoothing (Alon & Yahav, 2021; Li et al., 2018), and certain globally defined structures remain indistinguishable to 1–WL (Morris et al., 2020). In our setting, whether a link exists may hinge on distant common ancestors (shared base checkpoints or datasets), and the *type* of an edge may be resolved only by comparing signals spread across parallel provenance paths. To address these non-local requirements, we complement GIN with a small number of Graph Transformer layers (Dwivedi & Bresson, 2021; Ying et al., 2021b) that perform global, content-aware routing of information. While global attention is not guaranteed to be injective in the 1–WL sense, it supplies the missing *global context*, enabling the model to aggregate evidence across arbitrary distances. This 1–WL-informed perspective thus motivates our hybrid: GIN layers secure strong local expressivity for both link prediction and edge typing, and GT layers supply the long-range reasoning that local refinement alone cannot provide.

## C  AI SUPPLY CHAIN GRAPH CONSTRUCTION

We construct the AI supply chain graph from the Hugging Face platform (Hugging Face) with the following four steps. The resulting graph contains a **single node type** (model repositories) and **four directed edge types**: *fine-tuned*, *adapted*, *quantized*, and *merged*. Below we provide details about

graph design, node and edge attributes, text embeddings, and incorporation of structured metadata and repository-level information.

(i) *Model tree extraction.* Platforms maintain a hierarchical, developer-declared "model tree" that records parent–child inheritance (e.g., a fine-tuned checkpoint pointing to its base). Because this information is user-supplied and often incomplete, we traverse each model's tree to recover declared ancestors/descendants and use these declarations to initialize directed edges. The top–down nature of these trees induces edge directionality from source/base to derived nodes. *For edge design*, we restrict the graph to a single node type (model). From the model tree and associated metadata, we construct four types of edges: (1) **fine-tuned** edges when the child is trained from a base model; (2) **adapted** edges when LoRA/adapter modules or PEFT configurations are present; (3) **quantized** edges when the child repository represents a low-bit or compressed version of an existing model; and (4) **merged** edges when a model declares merging multiple parents. Each edge stores attributes such as `relationship_type`, whether the source was *explicitly declared* or *weakly inferred*, and (for merges) the number of parents involved. All the edges point from the *source/base* model to the *derived* model, following the top-down structure of the declared tree.

(ii) *Model card analysis.* We parse model cards to harvest training objectives, declared bases/merges, datasets, quantization mentions, licenses, and citations. The free-text card is embedded into a fixed-length vector and attached to the node as part of its attributes. For the textual portion of the model card, we use a sentence-level transformer encoder *all-MiniLM-L6-v2* from SentenceTransformer) to embed the model card into a 384-dimensional representation. This family of encoders is chosen for its strong semantic similarity properties and robustness to long-form technical descriptions, making it well suited for capturing supply-chain information. The embedding is stored as a node attribute and is not used to create edges.

(iii) *Structured metadata.* From repository configuration files (e.g., `config.json`, `tokenizer.json`), we extract static descriptors, such as num_layers, hidden_size, num_heads, vocab_size, positional embedding type, activation, tokenizer type, special tokens, and normalization strategy.

(iv) *Repository scan and checkpoints.* We enumerate file types/sizes and detect formats (e.g., `*.safetensors`, `*.gguf` with bit-widths), adapters (e.g., `adapter_config.json`), and ONNX exports. We also compute lightweight weight summaries (parameter counts, mean/std of selected tensors, optional sparsity stats).

## D   IMPACT OF VARYING AI SUPPLY CHAIN GRAPH SIZE

To assess the robustness and scalability of CHAININFER, we evaluate performance on supply chain graphs of three different sizes: 10K, 100K, and 200K nodes. To construct smaller graphs, we randomly sample a subset of nodes from the full Hugging Face graph and then retain all edges whose endpoints lie within the selected node set. This procedure preserves local degree distributions, edge-label proportions, and structural motifs, ensuring that each subgraph remains a representative microcosm of the full dataset rather than an arbitrary fragment. By progressively enlarging the graph, we are able to examine how CHAININFER adapts to increasing structural complexity, label imbalance, and feature sparsity.

Figure 3 summarizes the results in terms of joint accuracy over training epochs. Several trends are evident. (i) On the smallest graph (10K nodes), CHAININFER converges rapidly within ∼100 epochs and achieves ≈92% joint accuracy, suggesting that the model is effective even on relatively modest datasets. (ii) On the medium-sized graph (100K nodes), convergence is slower, but the final accuracy still reaches ≈89%. This demonstrates resilience to larger search spaces and noisier metadata,

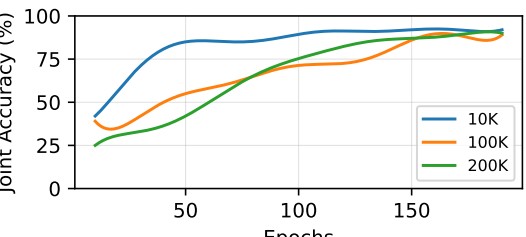

Figure 3: Convergence of CHAININFER for varying graph sizes.

though more training epochs are required. (iii) On the full 200K-node graph, accuracy initially lags due to the scale and sparsity of observed edges, but the model steadily improves and stabilizes

Table 6: Performance of with or without learning imputer (LI) under missing features.

| Missing features | Methods | $\text{Acc}_{\text{edge}}\uparrow$ | $\text{F1}_{\text{edge}}\uparrow$ | $\text{Auc}_{\text{edge}}\uparrow$ | $\text{Acc}_{\text{link}}\uparrow$ | $\text{F1}_{\text{link}}\uparrow$ | $\text{Auc}_{\text{link}}\uparrow$ | $\text{Acc}_{\text{joint}}\uparrow$ |
|---|---|---|---|---|---|---|---|---|
| 0% | Without LI | 0.94 | 0.93 | 0.96 | 0.91 | 0.94 | 0.94 | 0.94 |
| | with LI | 0.93 | 0.92 | 0.93 | 0.91 | 0.94 | 0.93 | 0.93 |
| 25% | Without LI | 0.88 | 0.87 | 0.89 | 0.90 | 0.91 | 0.93 | 0.91 |
| | with LI | 0.91 | 0.91 | 0.92 | 0.94 | 0.93 | 0.94 | 0.93 |
| 50% | Without LI | 0.81 | 0.82 | 0.84 | 0.84 | 0.83 | 0.86 | 0.84 |
| | with LI | 0.87 | 0.88 | 0.90 | 0.92 | 0.91 | 0.92 | 0.90 |
| 75% | Without LI | 0.71 | 0.72 | 0.75 | 0.75 | 0.74 | 0.75 | 0.73 |
| | with LI | 0.84 | 0.83 | 0.87 | 0.86 | 0.85 | 0.86 | 0.85 |
| 100% | Without LI | 0.43 | 0.41 | 0.43 | 0.49 | 0.50 | 0.55 | 0.41 |
| | with LI | 0.45 | 0.41 | 0.42 | 0.47 | 0.49 | 0.52 | 0.37 |

around 90%. Importantly, despite the increased difficulty, CHAININFER preserves near-parity with its performance on smaller graphs, indicating that it generalizes across scales rather than overfitting to a particular size.

# E  EFFECTIVENESS OF MISSINGNESS-AWARE LEARNED IMPUTER STRATEGY

Section 3.4 introduced the missingness-aware Learned Imputer (LI), which is designed to improve robustness when node metadata is incomplete—a common characteristic of real-world AI supply-chain graphs. To quantify its contribution, we conduct a controlled missingness study in which node features are progressively masked at rates of 0%, 25%, 50%, 75%, and 100%. We compare two variants of our model: one using the full Learned Imputer and one using a simple zero-filling baseline.

Table 6 summarizes the results across edge classification, link prediction, and joint performance. When no features are missing, the two variants achieve similar accuracy, as expected. However, as missingness increases, the Learned Imputer consistently yields stronger performance across all metrics. The improvements are especially pronounced at higher missingness levels. For example, at 75% missingness, the Learned Imputer improves $\text{F1}_{\text{edge}}$ from 0.72 to 0.83 and $\text{Acc}_{\text{joint}}$ from 0.73 to 0.85. These results demonstrate that conditioning on missingness and incorporating the mask into the encoder significantly enhances robustness to partially observed metadata.

At 100% missingness, both variants converge to similar performance, since no semantic information is available and the model must rely solely on structural context. This behavior aligns with expectations and further illustrates that the Learned Imputer provides substantial benefits whenever partial metadata is available, while remaining stable in the extreme case of fully missing features.

# F  GENERALIZING TO ADDITIONAL AI SUPPLY CHAIN ENTITIES

To assess the extensibility of CHAININFER beyond model–model provenance, we expand the supply chain graph to include dataset nodes, resulting in a heterogeneous graph comprising both models and datasets. This extension allows us to evaluate whether CHAININFER can infer relationships across different categories of supply chain entities.

**Heterogeneous graph construction.** We developed a specialized extraction pipeline to characterize dataset nodes, distinct from model nodes. For each dataset repository identified in the supply chain, we extract a multi-modal feature vector by processing two key files: (i) Semantic Representation: We download the README.md (Dataset Card) and encode the textual description using Sentence-Transformer(all-MiniLM-L6-v2). This captures the semantic domain of the data (e.g., "legal text", "medical imaging"). (ii) Structured Metadata: We parse the YAML frontmatter and dataset_info.json to extract explicit attributes, including task categories (e.g., question-answering), Languages (e.g.,

Table 7: Extending CHAININFER to heterogeneous graph.

| Node Type | $Acc_{edge}\uparrow$ | $F1_{edge}\uparrow$ | $AUC_{edge}\uparrow$ | $Acc_{link}\uparrow$ | $F1_{link}\uparrow$ | $AUC_{link}\uparrow$ | $Acc_{joint}\uparrow$ | $Time\downarrow$ |
|---|---|---|---|---|---|---|---|---|
| Model only | 0.94 | 0.93 | 0.96 | 0.91 | 0.94 | 0.94 | 0.94 | 2.34 |
| Model & dataset | 0.81 | 0.83 | 0.85 | 0.87 | 0.90 | 0.93 | 0.89 | 4.39 |

Table 8: Ablation study of different feature categories.

| Feature Removed | $Acc_{edge}\uparrow$ | $F1_{edge}\uparrow$ | $Auc_{edge}\uparrow$ | $Acc_{link}\uparrow$ | $F1_{link}\uparrow$ | $Auc_{link}\uparrow$ | $Acc_{joint}\uparrow$ |
|---|---|---|---|---|---|---|---|
| None | **0.94** | **0.93** | **0.96** | **0.91** | **0.94** | **0.94** | **0.94** |
| Architecture | 0.79 | 0.72 | 0.81 | 0.79 | 0.81 | 0.83 | 0.81 |
| Tokenizer & preprocessing | 0.84 | 0.81 | 0.81 | 0.84 | 0.81 | 0.84 | 0.86 |
| Quantization & pruning | 0.84 | 0.84 | 0.87 | 0.85 | 0.85 | 0.87 | 0.86 |
| File-level metadata | 0.80 | 0.81 | 0.84 | 0.83 | 0.84 | 0.83 | 0.83 |
| Model fingerprints | 0.84 | 0.84 | 0.86 | 0.84 | 0.87 | 0.87 | 0.86 |
| PEFT | 0.88 | 0.90 | 0.91 | 0.87 | 0.90 | 0.91 | 0.89 |

en, zh) and Size Categories. These semantic and structural features are concatenated to form the final node representation.

**Establishing Relationships.** We construct the dependency graph by parsing the provenance metadata of Model nodes. We specifically look for the datasets: tag in the dataset metadata. If dataset B lists model A in its metadata, we create a "trained_on" directed edge(B, A), which means model A use dataset B for training. This edge represents the flow of information from raw material to product. Unlike model-to-model lineage, this relationship captures the foundational knowledge injection into the supply chain.

**Joint Training Methodology.** We collected 46,213 dataset nodes and 58,761 "trained_on" edges. The full heterogeneous graph consists of 246,213 nodes and 251,777 edges. We treat the system as a Heterogeneous Graph Learning problem. We employ a coupled encoder where a GIN (Graph Isomorphism Network) layer aggregates local neighborhood information, followed by a Graph Transformer (GT) layer to capture global lineage dependencies. Optimization: The model is trained end-to-end on two tasks: Link Prediction and Edge Classification (inferring the type of relationship). The loss function sums the cross-entropy losses from both tasks, updating the weights of both Model and Dataset feature encoders simultaneously.

Table 7 compares the performance of CHAININFERon the full supply-chain graph (models + datasets) versus the model-lineage graph (models only). The results show that our method maintains high joint accuracy even in the heterogeneous setting that includes dataset nodes, indicating that it can still uncover meaningful relationships within a more complex, multi-type supply chain. However, we observe that the standalone classification performance on heterogeneous graphs is lower, which reduces the final joint accuracy. We attribute this to incomplete or low-quality feature extraction for dataset nodes in the current implementation. Improving dataset-node representations is therefore a key direction for future work.

## G    FEATURE CATEGORY ABLATION STUDY

To understand the relative contribution of different feature families, we conducted a feature category ablation study in which we systematically removed one category of features at a time during both training and testing. This experiment allows us to disentangle the importance of structural cues versus various content-level metadata. The full results are included in the appendix, with a summary shown in Table 8. We made three interesting findings.

(i) Different feature families contribute in distinct ways to inference accuracy. The most influential feature category is architecture-related information, whose removal causes the largest drop in joint accuracy (from 0.94 to 0.81). This suggests that architectural descriptors provide strong semantic signals for distinguishing transformation relationships among models.

Table 9: Joint accuracy under different edge noise levels.

| Noise level | 0% | 10% | 20% | 30% | 40% | 50% |
|---|---|---|---|---|---|---|
| **Joint accuracy** | 0.94 | 0.90 | 0.89 | 0.89 | 0.84 | 0.72 |

Table 10: Comparison between a text-based MLP method and CHAININFER.

| Method | $Acc_{edge}\uparrow$ | $F1_{edge}\uparrow$ | $Auc_{edge}\uparrow$ | $Acc_{link}\uparrow$ | $F1_{link}\uparrow$ | $Auc_{link}\uparrow$ | $Acc_{joint}\uparrow$ |
|---|---|---|---|---|---|---|---|
| MLP | 0.86 | 0.87 | 0.91 | 0.91 | 0.93 | 0.92 | 0.87 |
| CHAININFER | 0.94 | 0.93 | 0.96 | 0.91 | 0.94 | 0.94 | 0.94 |

(ii) Other content-level features, such as tokenizer and preprocessing metadata, quantization and pruning indicators, and file-level metadata, also contribute meaningfully, with accuracy decreases in the range of 0.08 to 0.11 when removed. These results demonstrate that semantic and configuration-specific cues are actively used by the model and are not overshadowed by structural information.

(iii) At the same time, the model remains reasonably stable when individual categories are removed, which suggests that structural signals and neighborhood aggregation compensate for partial feature loss. For instance, removing PEFT-related information or fingerprints produces only modest accuracy declines, indicating that the encoder can recover some of these semantics from neighboring nodes or lineage patterns.

## H ROBUSTNESS TO NOISY OR INCORRECT LINEAGE INPUTS

In real-world model repositories, metadata and lineage declarations may contain noise, inconsistencies, or erroneous parent–child relationships. Although our primary experiments assume clean inputs, the architecture of CHAININFERis inherently resilient to such imperfections because it aggregates information from semantic content and structural lineage patterns. When an incorrect lineage edge is introduced, semantic incompatibility between the involved models, together with disagreement from surrounding neighbors, enables the encoder to downweight the misleading signal.

To evaluate this robustness, we conduct a noise-injection study in which a proportion of edges in the test graph are randomly corrupted to simulate incorrect or adversarial lineage declarations. Table 9 reports joint accuracy under noise levels ranging from 0% to 50%. The model maintains high accuracy under moderate corruption, retaining 0.89 joint accuracy at 30% noise, indicating strong structural and semantic redundancy in the learned representations. However, at very high corruption levels (50%), performance degrades more substantially (0.72 joint accuracy), suggesting that extreme graph distortion can overwhelm the available contextual cues. These results highlight both the robustness of the current design and the importance of developing even stronger mechanisms for detecting and mitigating incorrect lineage information in future work.

## I COMPARISON WITH NON-GRAPH BASELINES

To complement the graph-based comparisons in the main paper, we additionally evaluate a strong non-graph baseline that relies purely on textual and metadata features. In particular, we implement a multi-layer perceptron (MLP) classifier trained on the node feature embeddings without access to any structural information. Similar text-driven classifiers have been widely used in prior supply chain analysis (Chen & Chen, 2025; Kalboussi et al., 2024), making the MLP an appropriate representative of non-graph approaches.

Table 10 shows the performance comparison. The MLP method achieves competitive performance on link prediction ($Acc_{link} = 0.91$), where simple textual similarity is often sufficient. However, the model performs substantially worse on edge classification ($Acc_{edge} = 0.86$ vs. 0.94 for CHAININFER) and on joint accuracy (0.87 vs. 0.94). These tasks require distinguishing fine-grained transformation types and capturing multi-hop structural dependencies, which are the critical information that text-only models cannot access.

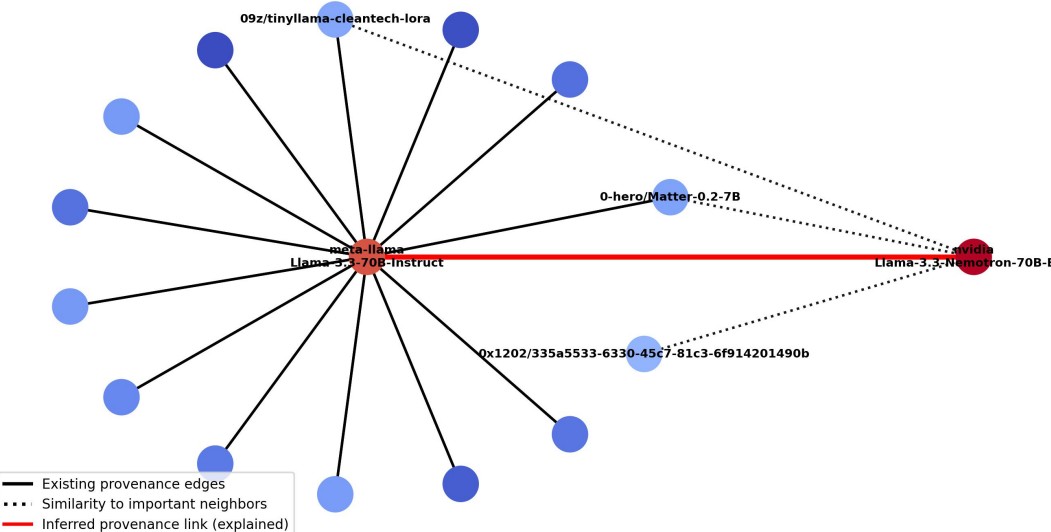

Figure 4: An example explanation generated by GNNExplainer for a fine-tune relation inferred by CHAININFER. Deeper colors mean the nodes are more important.

In contrast, CHAININFER integrates semantic metadata with both local message passing and global structural aggregation, enabling it to recover lineage and transformation patterns that are invisible to text-only classifiers. Overall, the comparison shows that while text-based methods provide a reasonable baseline, they are insufficient for full provenance inference. Incorporating structural information is essential for achieving high accuracy across all tasks, validating the need for a graph-based approach.

## J EXPLAINABILITY OF CHAININFER

To support transparent and auditable provenance reasoning, we integrate **GNNExplainer** (Ying et al., 2019) into CHAININFER. This enables users to inspect the structural and semantic evidence underlying each inferred relationship, which is essential in regulatory auditing and supply-chain risk assessment.

**Explainer Formulation.** Given a trained model $f(\cdot)$, a target node pair $(u, v)$, and the predicted relation $Y = f(G, u, v)$, GNNExplainer learns soft masks over edges and node features,

$$\mathbf{M}_E \in [0,1]^{|E|}, \qquad \mathbf{M}_X \in [0,1]^d,$$

which induce a perturbed subgraph $\hat{G}$ and perturbed features $\hat{X}$. The explainer optimizes these masks to maximize the mutual information between the prediction on the perturbed graph and the original model output:

$$\max_{\mathbf{M}_E, \mathbf{M}_X} I\Big(Y \,;\, f(\hat{G}, u, v)\Big).$$

This objective identifies the minimal structural and feature-level evidence required to reproduce the decision. Because CHAININFER performs both link prediction and edge-type classification jointly, we apply GNNExplainer directly to the joint prediction head. This ensures that the extracted explanations faithfully reflect the combined signals used during inference, without modifying or retraining the model.

**Case Study.** Figure 4 illustrates an example from our supply chain graph. We apply GNNExplainer to the model `nvidia/Llama-3.3-Nemotron-70B-Edit` to explain why CHAININFER predicts a *fine-tune* relation originating from `meta-llama/Llama-3.3-70B-Instruct`. The explainer highlights the key provenance scaffold supporting this prediction (shown as the solid red edge), while assigning low mask values to other neighboring connections (shown as dotted edges). This yields a concise, human-interpretable evidence subgraph, demonstrating that CHAININFER appropriately focuses on relevant lineage cues while suppressing spurious context.

