# OpenReview forum: "ChainInfer: A Joint Method for Inferring Missing AI Supply Chain Information"
_ICLR.cc/2026/Conference — ICLR 2026 Conference Desk Rejected Submission_

### Official Review · Reviewer_6s7p · 2025-10-30

**Soundness:** 3
**Presentation:** 3
**Contribution:** 3
**Rating:** 6
**Confidence:** 4

**Summary:**

This paper introduces CHAININFER, a joint graph learning framework for AI supply chain provenance inference. The authors formalize the problem as a coupled link prediction and edge classification task on large attributed graphs representing model dependencies in platforms such as Hugging Face and Kaggle. CHAININFER combines Graph Isomorphism Networks (GIN) for efficient local structural encoding and Graph Transformers (GT) for global context reasoning, trained end-to-end with a joint objective. Using a benchmark of 200K models, CHAININFER achieves a joint accuracy of 0.93 on Hugging Face and 0.90 in a zero-shot transfer to Kaggle, outperforming both GNN- and Transformer-based baselines. The paper demonstrates that hybrid architectures can accurately and efficiently infer missing metadata in large-scale AI ecosystems.

**Strengths:**

The paper clearly defines AI supply chain inference as a coupled graph task. Its key strength lies in the novel formulation of the problem as a coupled graph learning task, combining link prediction and edge classification in a unified framework. The proposed CHAININFER architecture integrates GIN-based local reasoning with Graph Transformer–based global context modeling. The validity of the model was verified through extensive experiments on large-scale real-world data, including 200K-node graphs from Hugging Face and zero-shot transfer to Kaggle. The work demonstrates 0.93 joint accuracy while maintaining computational efficiency. Moreover, CHAININFER generalizes inductively to Kaggle, retaining 0.90 accuracy without retraining.

**Weaknesses:**

The paper has a few limitations that could be addressed to strengthen its scope and generality. First, the current graph formulation focuses exclusively on model-level provenance, omitting other critical entities such as datasets, software components, and hardware, which limits the comprehensiveness of the inferred supply chain. Second, while the comparisons among GNN and Transformer variants are thorough, the absence of non-graph baselines—such as text-based metadata completion or knowledge graph embedding methods—makes it difficult to quantify CHAININFER’s advantage over broader families of approaches. Finally, the method relies on a carefully engineered feature pipeline derived from platform-specific metadata, which might reduce robustness in less structured or inconsistent repositories.

**Questions:**

1.	CHAININFER’s performance depends on extracted metadata features such as architecture parameters, tokenizer type, and quantization schemes. Could the authors provide a more systematic analysis of how performance degrades under varying degrees of metadata incompleteness or corruption? For instance, would the model still maintain high accuracy if 50% of metadata fields were missing or noisy?
2.	Real-world AI repositories often contain inconsistent or misleading metadata (e.g., false parent declarations or ambiguous model naming). How robust is CHAININFER to such noisy or adversarially incorrect inputs? Does the model include any mechanism (e.g., uncertainty estimation or confidence scoring) to detect or flag potentially unreliable inferred edges?
3.	The current formulation models only AI models as graph nodes, excluding datasets, software packages, and hardware dependencies that also constitute the AI supply chain. Do the authors foresee straightforward extensions of CHAININFER to heterogeneous graphs including multiple node and edge types? If so, what specific architectural modifications would be required?
4.	The feature extraction pipeline includes six distinct categories (architecture, tokenizer, quantization, metadata, fingerprints, and PEFT). Could the authors provide quantitative insights (e.g., via feature ablation or attention analysis) into which feature families contribute most strongly to inference accuracy? This would also clarify whether the model’s performance is dominated by structural signals or content-level cues.
5.	The paper compares CHAININFER only with graph-based models. Have the authors evaluated or considered baselines such as text-embedding–based metadata completion methods?
6.	In applied settings (e.g., regulatory auditing or risk assessment), understanding why a link or relation was inferred can be crucial. Does CHAININFER provide any interpretability mechanism (e.g., attention visualization or subgraph attribution) that helps users verify or justify the inferred provenance links?

---

> ### Author Response · Authors · 2025-11-25
> **Response to Reviewer 6s7p [1/4]**
>
> > 1. CHAININFER’s performance depends on extracted metadata features such as architecture parameters, tokenizer type, and quantization schemes. Could the authors provide a more systematic analysis of how performance degrades under varying degrees of metadata incompleteness or corruption? For instance, would the model still maintain high accuracy if 50% of metadata fields were missing or noisy?
>
> Response: Thank you for raising this concern. As the reviewer correctly noted, a key component of our approach is the missingness-aware Learned Imputer described in Section 3.4. To directly evaluate its benefit, we conducted a new experiment that compares our method with and without the Learned Imputer (LI) added as Appendix E. The new experiment focuses on nodes with incomplete metadata, which is the natural setting where imputation plays an important role.
>
> In this study, we progressively masked node features at rates of 0%, 25%, 50%, 75%, and 100%. We then evaluated both versions of the model on edge classification, link prediction, and joint performance. The full results are presented in the table below.
> | Missing Features | Methods     | Acc_edge ↑ | F1_edge ↑ | Auc_edge ↑ | Acc_link ↑ | F1_link ↑ | Auc_link ↑ | Acc_joint ↑ |
> |:-----------------|:------------|-----------:|----------:|-----------:|-----------:|----------:|-----------:|------------:|
> | 0%               | Without LI  |       0.94 |      0.93 |       0.96 |       0.91 |      0.94 |       0.94 |        0.94 |
> |                  | With LI     |       0.93 |      0.92 |       0.93 |       0.91 |      0.94 |       0.93 |        0.93 |
> | 25%              | Without LI  |       0.88 |      0.87 |       0.89 |       0.90 |      0.91 |       0.93 |        0.91 |
> |                  | With LI     |       0.91 |      0.91 |       0.92 |       0.94 |      0.93 |       0.94 |        0.93 |
> | 50%              | Without LI  |       0.81 |      0.82 |       0.84 |       0.84 |      0.83 |       0.86 |        0.84 |
> |                  | With LI     |       0.87 |      0.88 |       0.90 |       0.92 |      0.91 |       0.92 |        0.90 |
> | 75%              | Without LI  |       0.71 |      0.72 |       0.75 |       0.75 |      0.74 |       0.75 |        0.73 |
> |                  | With LI     |       0.84 |      0.83 |       0.87 |       0.86 |      0.85 |       0.86 |        0.85 |
> | 100%             | Without LI  |       0.43 |      0.41 |       0.43 |       0.49 |      0.50 |       0.55 |        0.41 |
> |                  | With LI     |       0.45 |      0.41 |       0.42 |       0.47 |      0.49 |       0.52 |        0.37 |
>
> The results show a clear pattern. When no features are missing, both methods perform similarly, which is expected because imputation has no effect in this case. However, as missingness increases, the Learned Imputer consistently provides stronger performance. The improvements are especially pronounced at 50% and 75% missingness. For example, at 75% missingness, the Learned Imputer improves F1_edge from 0.72 to 0.83 and improves Acc_joint from 0.73 to 0.85. These results demonstrate that explicitly modeling missingness and providing the mask to the encoder significantly increases robustness to incomplete metadata, which is a common property of real AI supply-chain graphs.
>
> At 100% missingness, the two methods perform similarly. This is expected because neither model has access to any observed features, and the task relies primarily on structural information. This further supports our design choice: the Learned Imputer offers substantial benefits when meaningful partial information is available, and it behaves reasonably when no information is present.

---

> ### Author Response · Authors · 2025-11-25
> **Response to Reviewer 6s7p [2/4]**
>
> > 2. Real-world AI repositories often contain inconsistent or misleading metadata (e.g., false parent declarations or ambiguous model naming). How robust is CHAININFER to such noisy or adversarially incorrect inputs? Does the model include any mechanism (e.g., uncertainty estimation or confidence scoring) to detect or flag potentially unreliable inferred edges?
>
> Response: Thank you for this insightful question. We fully agree that public AI repositories often contain inconsistent or misleading metadata, such as incorrect parent declarations or ambiguous model naming, and robustness to such noise is an important consideration for practical deployment. In the current version of the work, we do not explicitly model adversarial or intentionally incorrect metadata; however, the design of our architecture provides a natural degree of resilience to isolated errors.
>
> In particular, CHAININFER aggregates information from three complementary views: (i) semantic content extracted from model cards and repository descriptions, (ii) structural lineage derived from the larger provenance graph, and (iii) local topological neighborhoods processed through GIN-based message passing. An incorrect or misleading metadata entry affects only one of these channels, while the remaining signals typically remain consistent. For instance, if a model advertises a “false parent,” the semantic mismatch between the child and the incorrect parent, combined with the disagreement from structurally related neighbors, allows the encoder to downweight this inconsistent signal during aggregation.
>
> Experimental Validation.
> To evaluate this empirically, we conducted a noise-injection study, now included in Appendix Section H (“Robustness to Noisy or Incorrect Inputs”). We randomly corrupted a portion of edges in the test set to simulate false or misleading parent declarations, varying the corruption rate from 10 percent to 50 percent. The joint accuracy results are summarized in the below Table.
>
> | Noise level      | 0%   | 10%  | 20%  | 30%  | 40%  | 50%  |
> |:-----------------|-----:|-----:|-----:|-----:|-----:|-----:|
> | Joint accuracy   | 0.94 | 0.90 | 0.89 | 0.89 | 0.84 | 0.72 |
>
> The results indicate that CHAININFER maintains approximately 89% of its original accuracy with up to 30% corrupted edges. Accuracy begins to decline more noticeably at very high corruption levels (40–50%), which is expected when a large portion of the graph is adversarially perturbed.
>
> Overall, these findings suggest that CHAININFER is naturally robust to moderate amounts of inconsistent or noisy metadata. Developing explicit mechanisms for confidence scoring or uncertainty-aware inference remains an important direction for future work.
>
> >3. The current formulation models only AI models as graph nodes, excluding datasets, software packages, and hardware dependencies that also constitute the AI supply chain. Do the authors foresee straightforward extensions of CHAININFER to heterogeneous graphs including multiple node and edge types? If so, what specific architectural modifications would be required?
>
> Thanks for the question. Please refer to the "**Response to Reviewer KggE [2/2] Question 4**" Can the proposed CHAININFER framework be applied to infer the relationship between other elements of the AI supply chain?

---

> ### Author Response · Authors · 2025-11-25
> **Response to Reviewer 6s7p [3/4]**
>
> > 4. The feature extraction pipeline includes six distinct categories (architecture, tokenizer, quantization, metadata, fingerprints, and PEFT). Could the authors provide quantitative insights (e.g., via feature ablation or attention analysis) into which feature families contribute most strongly to inference accuracy? This would also clarify whether the model’s performance is dominated by structural signals or content-level cues.
>
> Response: Thank you for this insightful suggestion. To better understand them, we conducted a feature ablation study in which we removed each of the six feature categories during both training and evaluation. The full results have been added to Appendix (Feature Ablation Study).
>
> The following table summarized the results. Particularly, we made three interesting findings.
> | Feature Removed            | Acc_edge ↑ | F1_edge ↑ | AUC_edge ↑ | Acc_link ↑ | F1_link ↑ | AUC_link ↑ | Acc_joint ↑ |
> |:---------------------------|-----------:|----------:|-----------:|-----------:|----------:|-----------:|------------:|
> | None                       | **0.94**   | **0.93**  | **0.96**   | **0.91**   | **0.94**  | **0.94**   | **0.94**    |
> | Architecture               | 0.79       | 0.72      | 0.81       | 0.79       | 0.81      | 0.83       | 0.81        |
> | Tokenizer & preprocessing  | 0.84       | 0.81      | 0.81       | 0.84       | 0.81      | 0.84       | 0.86        |
> | Quantization & pruning     | 0.84       | 0.84      | 0.87       | 0.85       | 0.85      | 0.87       | 0.86        |
> | File-level metadata        | 0.80       | 0.81      | 0.84       | 0.83       | 0.84      | 0.83       | 0.83        |
> | Model fingerprints         | 0.84       | 0.84      | 0.86       | 0.84       | 0.87      | 0.87       | 0.86        |
> | PEFT                       | 0.88       | 0.90      | 0.91       | 0.87       | 0.90      | 0.91       | 0.89        |
>
>  (1) Different feature families contribute in distinct ways to inference accuracy. The most influential feature category is architecture-related information, whose removal causes the largest drop in joint accuracy (from 0.94 to 0.81). This suggests that architectural descriptors provide strong semantic signals for distinguishing transformation relationships among models.
>
> (2) Other content-level features, such as tokenizer and preprocessing metadata, quantization and pruning indicators, and file-level metadata, also contribute meaningfully, with accuracy decreases in the range of 0.08 to 0.11 when removed. These results demonstrate that semantic and configuration-specific cues are actively used by the model and are not overshadowed by structural information.
>
> (3) The model remains reasonably stable when individual categories are removed, which suggests that structural signals and neighborhood aggregation compensate for partial feature loss. For instance, removing PEFT-related information or fingerprints produces only modest accuracy declines, indicating that the encoder can recover some of these semantics from neighboring nodes or lineage patterns.
>
> > 5. The paper compares CHAININFER only with graph-based models. Have the authors evaluated or considered baselines such as text-embedding–based metadata completion methods?
>
> Response: Thank you for this helpful comment. To address this concern, we added a new experiment to compare with a strong non-graph baseline that relies purely on textual and metadata features. Specifically, we trained a multi-layer perceptron (MLP) classifier on the node feature embeddings, without using any structural information. The results are shown in the below Table. This experiment has been added to Appendix Section I “Comparison with Non-Graph Baselines”.
>
> | Method      | Acc_edge ↑ | F1_edge ↑ | AUC_edge ↑ | Acc_link ↑ | F1_link ↑ | AUC_link ↑ | Acc_joint ↑ |
> |:------------|-----------:|----------:|-----------:|-----------:|----------:|-----------:|------------:|
> | MLP         |       0.86 |      0.87 |       0.91 |       0.91 |      0.93 |       0.92 |        0.87 |
> | CHAININFER  |       0.94 |      0.93 |       0.96 |       0.91 |      0.94 |       0.94 |        0.94 |
>
> The MLP performs reasonably well on link prediction (0.91 accuracy), which is expected because surface-level textual similarity often provides useful cues for identifying whether two models may be related. However, the MLP performs substantially worse on edge classification (0.86 accuracy compared to 0.94 for CHAININFER) and on joint accuracy (0.87 compared to 0.94). These tasks require distinguishing fine-grained transformation types and leveraging multi-hop structural dependencies—information that a text-only model cannot access. In contrast, CHAININFER combines semantic metadata with both local neighborhood reasoning and global structural aggregation, enabling it to capture lineage patterns and transformation semantics that are invisible to text-only approaches.

---

> ### Author Response · Authors · 2025-11-25
> **Response to Reviewer 6s7p [4/4]**
>
> > 6. In applied settings (e.g., regulatory auditing or risk assessment), understanding why a link or relation was inferred can be crucial. Does CHAININFER provide any interpretability mechanism (e.g., attention visualization or subgraph attribution) that helps users verify or justify the inferred provenance links?
>
> Response: We thank the reviewer for emphasizing the importance of explainability. To support verifiable provenance reasoning, we integrated GNNExplainer (Ying et al., NeurIPS 2019) into CHAININFER and added a new experiment to Appendix.
>
> GNNExplainer provides subgraph-level attribution by learning a soft mask over edges and node features. The mask is optimized to maximize the mutual information between the model’s prediction Y and the distribution of the masked subgraph G, thereby isolating the minimal set of structural and semantic factors necessary to reproduce the decision. This allows the explainer to highlight the true causal evidence while filtering out irrelevant or noisy connections.
>
> We conducted a qualitative case study on a real-world node (nvidia/Llama-3.3-Nemotron-70B-Edit) from our supply-chain graph. As shown in the figure, the explainer correctly surfaced the provenance scaffold supporting the inferred edge to the true parent model (meta-llama/Llama-3.3-70B-Instruct, solid red). In contrast, other potential neighbors receive only dotted, low-importance attributions, indicating that CHAININFER suppresses spurious connections. This yields a concise, human-interpretable evidence graph that auditors can use to justify and validate inferred lineage links. Please refer to the paper for Figure 4.

---

> > ### Comment · Reviewer_6s7p · 2025-11-26
> >
> > The authors have addressed my concerns. My general assessment of the submissions remains unchanged and thus I maintain my original score.

---

### Official Review · Reviewer_KggE · 2025-11-01

**Soundness:** 3
**Presentation:** 3
**Contribution:** 3
**Rating:** 6
**Confidence:** 2

**Summary:**

In the modern Artificial Intelligence (AI) ecosystem, the AI supply chain is used to define the entire ecosystem of data, models, hardware, and software required to develop, deploy, and maintain AI systems. However, incomplete metadata is often present in the AI supply chain. This paper focuses on the model provenance inference problem in this area and proposes CHAININFER, a joint method that exploits both Graph Neural Network (GNN) and Graph Transformer (GP) to do the link prediction and dependency type classification between models. The proposed hybrid modeling method is demonstrated to outperform other tested baselines in both model provenance inference accuracy and generalization ability.

**Strengths:**

1. This paper has tackled an important problem in constructing the modern AI ecosystem, i.e., model provenance. The related works are comprehensively analyzed, and the work is well motivated.

2. The joint model of GNN and GT is technically sound and demonstrated to achieve good performance in the real-world AI platforms (i.e., Hugging Face and Kaggle datasets).

**Weaknesses:**

1. The title and the introduction somewhat overclaim the contribution of this work. This paper claims to tackle the AI supply chain inference. I quota "we formalize AI supply chain inference as a coupled graph learning problem". However, only model provenance inference is considered in this work.

2. In the experiments, only the graph-based model provenance method is compared. Other types of approaches, such as those mentioned in the related work, are not empirically compared.

3. There exist some unclear claims and confusing notations. For example,

* In line 75, it's not clear what the "most existing approaches" refer to. Proper citations are needed.
* It's confusing how the inequality in line 283 is achieved, how (7) is achieved, and whether they are related.

**Questions:**

1. Can the proposed CHAININFER framework be applied to infer the relationship between other elements of the AI supply chain?

---

> ### Author Response · Authors · 2025-11-25
> **Response to Reviewer KggE [1/2]**
>
> > 1. The title and the introduction somewhat overclaim the contribution of this work. This paper claims to tackle the AI supply chain inference. I quota "we formalize AI supply chain inference as a coupled graph learning problem". However, only model provenance inference is considered in this work.
>
> Response: We appreciate the reviewer for pointing this out. We agree that the "AI Supply Chain" encompasses a broad spectrum of components (e.g., dataset, software, and hardware) and that our work specifically targets the provenance and lineage aspect of this chain. As we cannot change the paper title anymore, we added a new paragraph to clearly define the scope at the end of the Introduction, which is listed below as well.
>
> “Scope. This work focuses exclusively on model-level provenance inference, which we identify as a core and tractable component of the broader AI supply chain. ChainInfer aims to recover missing or latent relationships among models, including fine-tuning, merging, quantization, and other transformation-induced dependencies, since these relationships form the backbone of provenance flows in modern model ecosystems. Other AI supply chain elements, including datasets, software libraries, training pipelines, and hardware environments, are outside the current scope due to the lack of standardized metadata. Nonetheless, ChainInfer establishes a foundation for scalable, graph-based inference methodologies that can be extended in future work to incorporate non-model entities, enabling a more comprehensive and end-to-end AI supply chain analysis.”
>
> >2. In the experiments, only the graph-based model provenance method is compared. Other types of approaches, such as those mentioned in the related work, are not empirically compared.
>
> Response: Thank you for this helpful comment. To address this concern, we added a new experiment to compare with a strong non-graph baseline that relies purely on textual and metadata features. Specifically, we trained a multi-layer perceptron (MLP) classifier on the node feature embeddings, without using any structural information. The results are shown in the below Table. This experiment has been added to Appendix Section I “Comparison with Non-Graph Baselines”.
>
> | Method      | Acc_edge ↑ | F1_edge ↑ | AUC_edge ↑ | Acc_link ↑ | F1_link ↑ | AUC_link ↑ | Acc_joint ↑ |
> |:------------|-----------:|----------:|-----------:|-----------:|----------:|-----------:|------------:|
> | MLP         |       0.86 |      0.87 |       0.91 |       0.91 |      0.93 |       0.92 |        0.87 |
> | CHAININFER  |       0.94 |      0.93 |       0.96 |       0.91 |      0.94 |       0.94 |        0.94 |
>
> The MLP performs reasonably well on link prediction (0.91 accuracy), which is expected because surface-level textual similarity often provides useful cues for identifying whether two models may be related. However, the MLP performs substantially worse on edge classification (0.86 accuracy compared to 0.94 for CHAININFER) and on joint accuracy (0.87 compared to 0.94). These tasks require distinguishing fine-grained transformation types and leveraging multi-hop structural dependencies—information that a text-only model cannot access.
>
> In contrast, CHAININFER combines semantic metadata with both local neighborhood reasoning and global structural aggregation, enabling it to capture lineage patterns and transformation semantics that are invisible to text-only approaches. Overall, the results show that non-graph methods are insufficient for full provenance inference, and that incorporating structural information is essential for achieving strong performance across all tasks.
>
> > 3. There exist some unclear claims and confusing notations. For example, in line 75, it's not clear what the "most existing approaches" refer to. Proper citations are needed. It's confusing how the inequality in line 283 is achieved, how (7) is achieved, and whether they are related.
>
> Response: We added citations for the “existing approaches”. We thank the reviewer for pointing out the ambiguity in our complexity explanation. The key idea is A **GT-only encoder** must stack many full-attention layers to capture both local and global dependencies, leading to a dominant cost of  **O(L\* · N² · d\*)**,  where *L\** and *d\** are the depth and hidden dimension needed to achieve the target accuracy. In our **hybrid architecture**, the inexpensive **GIN layers** first resolve all local interactions. This allows the GT module to use only a few much lighter layers, reducing the attention cost to **O(L_GT · N² · d_h)** with **L_GT ≪ L\***. The GIN part contributes only **O(L_GIN · M · d_h)**, which is linear in the number of edges *M*. The final **O(N · d_h²)** term corresponds to the standard MLP/readout and is included only for completeness.

---

> ### Author Response · Authors · 2025-11-25
> **Response to Reviewer KggE [2/2]**
>
> > 4. Can the proposed CHAININFER framework be applied to infer the relationship between other elements of the AI supply chain
>
> Response: We thank the reviewer for this valuable comment. Inspired by it, we have added a new experiment to study ChainInfer for other elements in the AI supply chain. Particularly, we added dataset nodes to our current graph, making it a heterogeneous graph with both model and dataset nodes. We detailed the construction, topology, and training process below. In addition, this has been added as a subsection in Appendix Section F Generalizing to Additional AI Supply Chain Entities.
>
> **1. Dataset Feature Collection**
>
> We developed a specialized extraction pipeline to characterize dataset nodes, distinct from model nodes. For each dataset repository identified in the supply chain, we extract a multi-modal feature vector by processing two key files: (i) Semantic Representation: We download the README.md (Dataset Card) and encode the textual description using Sentence-Transformer(all-MiniLM-L6-v2). This captures the semantic domain of the data (e.g., "legal text", "medical imaging"). (ii) Structured Metadata: We parse the YAML frontmatter and dataset_info.json to extract explicit attributes, including task categories (e.g., question-answering), Languages (e.g., en, zh) and Size Categories. These semantic and structural features are concatenated to form the final node representation.
>
> **2. Establishing Relationships (The Supply Chain Topology)**
>
> We construct the dependency graph by parsing the provenance metadata of Model nodes. We specifically look for the datasets: tag in the dataset metadata. If dataset B lists model A in its metadata, we create a “trained_on” directed edge(B,A), which means model A use dataset B for training. This edge represents the flow of information from raw material to product. Unlike model-to-model lineage, this relationship captures the foundational knowledge injection into the supply chain.
>
> **3. Graph Statistics.**
>
>  We collected 46213 dataset nodes and 58761 “trained_on” edges. The full heterogeneous graph consists of 246213 nodes and 251,777 edges.
>
> **4. Joint Training Methodology.**
>
> We treat the system as a Heterogeneous Graph Learning problem. Encoder: We employ a coupled encoder where a GIN (Graph Isomorphism Network) layer aggregates local neighborhood information (allowing Models to aggregate features from their parent Datasets), followed by a Graph Transformer (GT) layer to capture global lineage dependencies. Optimization: The model is trained end-to-end on two tasks: Link Prediction  and Edge Classification (inferring the type of relationship). The loss function sums the cross-entropy losses from both tasks, updating the weights of both Model and Dataset feature encoders simultaneously.
>
> **5. Evaluation.** We compared the performance of our Full Supply Chain Graph (Models + Datasets) against a baseline Model-Lineage Graph (Models only). The following table compares the performance of CHAININFER on the full supply-chain graph (models + datasets) versus the model-lineage graph (models only). The results show that our method maintains high joint accuracy even in the heterogeneous setting that includes dataset nodes, indicating that it can still uncover meaningful relationships within a more complex, multi-type supply chain. However, we observe that the standalone classification performance on heterogeneous graphs is lower, which reduces the final joint accuracy. We attribute this to incomplete or low-quality feature extraction for dataset nodes in the current implementation. Improving dataset-node representations is therefore a key direction for future work.
> | Node Type        | Acc_edge ↑ | F1_edge ↑ | AUC_edge ↑ | Acc_link ↑ | F1_link ↑ | AUC_link ↑ | Acc_joint ↑ | Time ↓ |
> |:-----------------|-----------:|----------:|-----------:|-----------:|----------:|-----------:|------------:|-------:|
> | Model only       |       0.94 |      0.93 |       0.96 |       0.91 |      0.94 |       0.94 |        0.94 |   2.34 |
> | Model & dataset  |       0.81 |      0.83 |       0.85 |       0.87 |      0.90 |       0.93 |        0.89 |   4.39 |

---

> > ### Comment · Reviewer_KggE · 2025-11-28
> >
> > I thank the authors for their detailed responses. My concerns have been addressed, and thus I will maintain my positive score.

---

### Official Review · Reviewer_7JT6 · 2025-11-01

**Soundness:** 3
**Presentation:** 3
**Contribution:** 3
**Rating:** 8
**Confidence:** 4

**Summary:**

The paper introduces a novel model for inferring missing AI supply chain information by framing it as a coupled graph learning problem: link prediction to recover missing dependencies and edge classification to determine their semantic types. The authors run several experiments across two datasets: a benchmark of 200K models from Hugging Face and another from Kaggle, demonstrating the model's ability to generalize inductively. The model achieves SOTA performance compared to several strong baselines.

**Strengths:**

The authors introduce a novel model that infers missing AI supply chain information and achieves SOTA performance compared with strong baselines across two datasets (Hugging Face and Kaggle). The authors provide a detailed rationale for the proposed model and the decisions made to address issues such as cold start and missing data. Furthermore, they conducted multiple experiments and ablation studies to demonstrate that the proposed model surpasses strong baselines across various configurations and to provide deeper insight into which architectural components and structural features drive performance. Their contribution is not limited to the model but also includes the creation of two datasets for the AI supply chain, which they promise to share upon acceptance. The paper is well-structured and clearly written. The proposed method will likely have a significant impact, given it can be applied not only to the AI supply chain but to broader software supply chains and even to supply chains not related to software.

**Weaknesses:**

We consider that the paper does not have many weaknesses. Nevertheless, we have identified two that we consider relevant:

 - While the authors describe the graph construction in the Appendix, there are some significant details missing. In particular, the authors provide insights into the information considered (e.g., Table 1), but no details about how such information is incorporated into the graph. How such information is modeled into a graph conditions how models are able to learn and their performance.

- We consider the related work exposed in the introduction to be of good quality. Nevertheless, we miss some brief paragraph highlighting models and approaches that have already been used for inferring missing AI supply chain information or for inferring missing supply chain information (e.g, in the context of software packages or other kinds of supply chains that pose similar challenges). The proposed baseline models are not grounded in prior work addressing this specific domain.

**Questions:**

1- While the authors describe the graph construction in the Appendix, we find certain details to be missing. We encourage the authors to provide information about (i) the graph design: in the manuscript is mentioned that different kind of edges exist: (a) what information is taken into account to build them?, (b) are they attributed?, (c) do attributes vary based on the kind of edge?, (d) do the authors consider different kind of nodes?, (ii) the authors mention using text embeddings: (a) what kind of models were used for such embeddings? (b) was there some specific rationale behind the choice?, (iii) the authors mention using structured metadata and performing a repository scan and checkpoints: (a) how is this information processed and incorporated into the graph? (b) are the metrics considered for enriching nodes or edges?. (iv) We encourage the authors to provide two Figures: (a) one detailing the graph construction process and (b) one describing the graph structure.

 2- We consider the related work exposed in the introduction to be of good quality. Nevertheless, we miss some brief paragraph highlighting models and approaches that have already been used for inferring missing AI supply chain information or for inferring missing supply chain information (e.g,. in the context of software packages or other kind of supply chains that pose similar challenges). We would appreciate the authors then ground the selected baselines in such works.

 3- While the authors report on the performance and time (seconds per epoch), they do not provide insights into the hardware that was used to execute the experiments. We encourage them to include in the Appendix a brief description of the hardware configuration used to run the experiments.

 4- All tables: ensure the numbers are aligned to the right so that differences in magnitude become evident

 5- Table 2: bolded results mean the best ones?

 6- Table 3, 4, 5: highlight best results.

 7- We encourage the authors to test whether the difference in performance noticed among the best models is statistically significant.

---

> ### Author Response · Authors · 2025-11-25
> **Response to Reviewer 7JT6 [1/2]**
>
> >  1. While the authors describe the graph construction in the Appendix, we find certain details to be missing. We encourage the authors to provide information about (i) the graph design: in the manuscript is mentioned that different kind of edges exist: (a) what information is taken into account to build them?, (b) are they attributed?, (c) do attributes vary based on the kind of edge?, (d) do the authors consider different kind of nodes?, (ii) the authors mention using text embeddings: (a) what kind of models were used for such embeddings? (b) was there some specific rationale behind the choice?, (iii) the authors mention using structured metadata and performing a repository scan and checkpoints: (a) how is this information processed and incorporated into the graph? (b) are the metrics considered for enriching nodes or edges?. (iv) We encourage the authors to provide two Figures: (a) one detailing the graph construction process and (b) one describing the graph structure.
>
> We thank the reviewer for the valuable suggestions to clarify the graph construction process. We have expanded the graph construction section (Appendix C) to explicitly address these points. Below, we provide the detailed answers corresponding to the reviewer's questions.
>
> **1.Graph Design.** (i) Node/Edge Construction: The graph is constructed by reconstructing the model tree (lineage) for each model. We treat each model as a parent node and identify its children by parsing the model tree URLs. We create directed edges to represent these dependencies. This process effectively maps the "supply chain" of the model, resulting in a directed acyclic graph (DAG) where edges (u,v) signify that model v depends on model u. (ii) Edge Attributes: Edges are categorized by their relational role in this tree (e.g., fine-tuned, quantized, adapted, and merged). Currently, we use these types as structural priors for the GNN message passing rather than attaching continuous feature vectors to edges. (iii) Node type: Currently, the graph is homogeneous with only model nodes. However, it can be easily extended to include other types of nodes, such as, dataset, and software library. In fact, per other reviewers request, we have added a new experiment with one more node type, i.e., dataset node. For more information, please see the response to Reviewer 4 Question 3.
>
> **2.Text Embeddings.** We use Sentence-Transformer (all-MiniLM-L6-v2) to encode the textual descriptions. This model was chosen for its efficiency in handling the large-scale, noisy text typical of model cards while preserving semantic clustering better than keyword matching. After getting the 384-dimensional output, we use PCA to reduce the dimension to 32, and then concatenate with other features.
>
> **3.Metadata Processing.** (i) Processing Pipeline: We implemented a multi-stage scraping pipeline that targets specific files within a repository to construct the node feature vector. (ii) Model Architecture: We parse config.json to extract hyperparameters such as num_hidden_layers, hidden_size, num_attention_heads, and vocab_size. (iii) Weight Fingerprinting: Uniquely, we inspect .safetensors files (if present). We load a sample of the model's tensors (up to 20) to calculate statistical moments (mean and standard deviation of the weights). This provides a continuous "fingerprint" of the model's weight distribution without loading the full model. (iv) Quantization & GGUF: We detect quantization status by checking for .gguf files or "quant" keywords in filenames, and parse GGUF headers for low-level metadata. (v) Heuristic Parsing: We use regular expressions on the README.md to extract parameter counts (e.g., "7B", "13M") if they are missing from the config. (vi) All extracted metrics are normalized and concatenated together to form the final node feature vector.

---

> ### Author Response · Authors · 2025-11-25
> **Response to Reviewer 7JT6 [2/2]**
>
> > 2. We consider the related work exposed in the introduction to be of good quality. Nevertheless, we miss some brief paragraph highlighting models and approaches that have already been used for inferring missing AI supply chain information or for inferring missing supply chain information (e.g,. in the context of software packages or other kind of supply chains that pose similar challenges). We would appreciate the authors then ground the selected baselines in such works.
>
> Response: Thank you for this insightful suggestion. We agree that connecting our baseline choices to existing work on supply chain inference strengthens the contextual grounding of our study. In the revised manuscript, we have added a new paragraph to the Related Work section (last paragraph) that highlights prior models and approaches used for inferring missing information in software ecosystems and knowledge graph settings. These domains share similar challenges with our task, including recovering undocumented dependencies, modeling incomplete provenance, and predicting latent relationships among interconnected artifacts.
>
> The added paragraph synthesizes relevant literature on software supply chain and knowledge graph completion. It also clarifies that models such as GCN, GAT, and GIN are widely used for link prediction and structural inference in these areas, which naturally motivates their use as baselines in our evaluation. Below is the added paragraph in our paper.
>
> “Missing metadata inference in other domains. Although the AI supply chain is an emerging area, the task of inferring missing provenance has clear parallels in software engineering and knowledge graph research. In software ecosystems such as npm, PyPI, and Software Heritage, software artifacts are modeled as dependency graphs, and prior work has investigated recovering missing or undocumented dependencies through static analysis, collaborative filtering, or graph-based representation learning Zhu & Zimmermann (2018); Mirhosseini & Parnin (2017); Abate et al. (2020). Related efforts in software supply chain security have framed dependency prediction and vulnerability propagation as link prediction problems on large heterogeneous graphs Zimmermann et al. (2019); Pashchenko et al. (2020). More broadly, provenance inference in interconnected systems has been studied within the knowledge graph completion literature, where models such as GCN, GAT, and GIN are widely used for relational link prediction and structural inference Kipf & Welling (2017); Velickovic et al. (2018); Xu et al. (2019b). These research directions collectively demonstrate that inferring missing connections in complex artifact graphs is commonly approached through graph completion or link prediction techniques across several domains.”
>
> > 3. While the authors report on the performance and time (seconds per epoch), they do not provide insights into the hardware that was used to execute the experiments. We encourage them to include in the Appendix a brief description of the hardware configuration used to run the experiments.
>
> Response: We thank the reviewer for noting this omission. In the revised manuscript, we have revised the first paragraph in Section 4.1 (Experimental Setup) to clearly explain them.
>
> “We implement CHAININFER using PyTorch Geometric (PyG) (Fey & Lenssen, 2019) and train all models on our in-house server. The server is equipped with four NVIDIA L40S GPUs, each with 40 GB of memory, although only a single GPU is used for each experiment. The system contains two AMD EPYC 9334 processors, each with 32 cores and hyperthreading enabled, for a total of 128 logical threads. All models are implemented in PyTorch 2.0.1 with CUDA 11.8, and we use PyG 2.6.1 for graph operations. Unless otherwise noted, all baselines are implemented within the same framework and are trained under identical hardware and optimizer settings to ensure fair comparison.”
>
> > 4. All tables: ensure the numbers are aligned to the right so that differences in magnitude become evident
>
> Thank you! We have revised all the Tables accordingly in the paper.
>
> > 5. Table 2: bolded results mean the best ones?
>
> Yes. We have explicitly clarified that for all the Tables in the paper.
>
> > 6. Table 3, 4, 5: highlight best results.
>
> Thanks. We have revised all of them in the paper accordingly.
>
> >7. We encourage the authors to test whether the difference in performance noticed among the best models is statistically significant.
>
> Thank you for this valuable suggestion. We have updated Table 2 (Performance and Time Comparison) in the paper to report the mean and standard deviation across 10 runs. The updated results show that CHAININFER consistently achieves the highest performance among all joint inference models. For all the other tables, we will add error bars in the final version.

---

> > ### Comment · Reviewer_7JT6 · 2025-11-28
> >
> > I thank the authors for their detailed responses. My concerns have been addressed, and thus I will maintain my positive score.

---

### Official Review · Reviewer_iqPC · 2025-11-01

**Soundness:** 2
**Presentation:** 2
**Contribution:** 2
**Rating:** 2
**Confidence:** 3

**Summary:**

The paper introduces a hybrid local GNN+ global Graph-Transformer method for recovering and classifying missing provenance links in AI supply chains. It treats the problem as link prediction + edge classification on large attributed graphs derived from Hugging Face, reporting strong but rather incremental performance against baselines and promising zero-shot generalization to Kaggle. Experiments compare against GNN-only, GT-only, and late-fusion baselines, showing performance gains and providing evidence for the considered principles.

**Strengths:**

1. Interesting and novel problem. The real world version of this corresponds to python package version conflicts for instance and could be potentially useful to resolve or accelerate.
2. An interesting dataset is proposed. I am not sure how novel this dataset is but but seems less useful compared to the problems discussed in the intro
3. Seemingly thorough evals and hyperparameter tuning results are reported.
4. Zero-shot generalization to new graphs is interesting.

**Weaknesses:**

1. Limited novelty in the proposed architecture. Local GNN + Global Attention variants have been tried before for the reasons outlined by the authors in the paper. See for instance Graph GPS by [Rampášek et. al. 2022](https://arxiv.org/pdf/2205.12454).
2. Line 386. "Joint" accuracy is a meaningless metric and should just be called accuracy. Edge existence can be reported separately (binary classification) and accuracy of classification is just accuracy. Seems like unnecessary obfuscation to me.
3. Evals are only on the newly constructed unbalanced provenance supply chain dataset with only 4 labels. This is a relatively easy setup where a chance model can achieve 25% classification accuracy.
4. The labels are also highly imbalanced. There are effectively 3 edge classes since one of them is super rare (<1%).
5. No results for the imputer strategy proposed in section 3.4
6. Improvements are incremental over GCN+Graphormer models in table 2 and likely statistically insignificant (no error bars are).

**Questions:**

See above weaknesses.

---

> ### Author Response · Authors · 2025-11-25
> **Response to Reviewer iqPC [1/3]**
>
> >1. Limited novelty in the proposed architecture. Local GNN + Global Attention variants have been tried before for the reasons outlined by the authors in the paper. See for instance Graph GPS by Rampášek et. al. 2022.
>
> Response: We appreciate the reviewer's comment and agree that combining local GNNs with global attention has appeared in prior work such as GraphGPS, Sgformer, and Exphormer. Our contribution is not the high-level pattern itself, but its problem-specific adaptation to the unique demands of AI supply chain provenance inference.
>
> First, our task setting is fundamentally different. We address a multi-task inference problem (joint link prediction and edge-type classification) on large, noisy, and incomplete heterogeneous supply-chain graphs. Prior works focus on single-task node-level or graph-level prediction and typically assume clean structural information.
>
> Second, we explicitly separate GIN layers and Graph Transformer layers. This design allows GIN to specialize in local provenance cues such as tokenizer overlap, adapter usage, and quantization artifacts, while the Graph Transformer focuses on long-range, multi-hop lineage reasoning. This separation is important because a unified stack often struggles to learn both fine-grained local inconsistencies and global structural dependencies at the same time. Existing local-plus-global models were developed for generic graph learning and do not consider this asymmetry.
>
> Finally, CHAININFER integrates a missingness-aware Learned Imputer to address the fact that more than 20 percent of nodes in the supply chain have incomplete metadata. This capability is not supported in previous architectures.
>
> >2. Line 386. "Joint" accuracy is a meaningless metric and should just be called accuracy. Edge existence can be reported separately (binary classification) and accuracy of classification is just accuracy. Seems like unnecessary obfuscation to me.
>
> Response: Sorry for the confusion. We truly appreciate the reviewer for pointing out this issue. In our experiment setting, there are actually three accuracy metrics as listed below.
>
> 1. Link prediction (binary classification)
> 2. Edge classification (multi-class classification)
> 3. Joint (link prediction + edge classification)
>
> Particularly, for edge classification accuracy (2), it is separated from link prediction (1) in our experiment. That is, these two metrics are computed independently on the same test set because they measure two separate capabilities.
>
> Differently, the joint accuracy (3) takes the results from both link prediction and edge classification into consideration. It is an end-to-end metric that uses the test set only once and counts an edge as correct only if the model both predicts that the edge exists and assigns the correct class. Thus, joint accuracy is strictly more demanding than either individual metric, as any error in link prediction or type prediction results in failure. This metric best reflects real-world usage, where a system must detect a relationship and correctly classify its type simultaneously.
>
> We have revised our paper accordingly to clearly discuss the definitions and differences between these three metrics (Section 4.1).
>
> > 3. Evals are only on the newly constructed unbalanced provenance supply chain dataset with only 4 labels. This is a relatively easy setup where a chance model can achieve 25% classification accuracy.
>
> **3.1 Regarding the newly constructed dataset.**
> In fact, our dataset is not arbitrary. It extends the public HuggingGraph dataset [CIKM 2025, arXiv version available since July 17, 2025, https://arxiv.org/abs/2507.14240v1] by adding node-level features that are essential for provenance inference.
>
> **3.2 Regarding unbalanced label distribution.**
> Thank you for the comment. Although the dataset includes four edge types, the problem is significantly more difficult than a balanced four-class classification task.
>
> **(1) The imbalance is inherent to real AI supply chains.**  Fine-tuning is by far the dominant transformation in ecosystems like Hugging Face and Kaggle, while adapt, quantize, and merge operations are naturally rare. Our label skew reflects actual provenance patterns rather than an artificial design choice, and accurately identifying these rare but important relations is part of the challenge.
>
> **(2) Inference occurs under noisy and incomplete information.** Provenance metadata is frequently missing, inconsistent, or ambiguous, and many true edges do not appear in the observable graph. The model must infer semantics from partial lineage chains and weak textual cues. This makes the task substantially harder than a standard four-class classifier with clean inputs.
>
> **(3) The evaluation is joint link prediction plus edge-type classification.** The model must first infer whether a link should exist at all and then determine its type. These two decisions are tightly coupled and far more challenging than classifying a known edge.

---

> ### Author Response · Authors · 2025-11-25
> **Response to Reviewer iqPC [2/3]**
>
> >4. The labels are also highly imbalanced. There are effectively 3 edge classes since one of them is super rare (<1%).
>
> Response: Thank you for pointing this out. We fully acknowledge that one of the edge types occurs with less than one percent frequency. This extreme rarity is an inherent property of real-world model ecosystems, where certain transformation types such as merging occur only in very specialized contexts. As a result, the dataset effectively contains three moderately frequent classes and one rare class.
>
> Although the rare class has limited representation, we include it for two reasons. First, it reflects genuine provenance behavior that should be modeled rather than removed, since the goal is to faithfully reconstruct real AI supply chains. Second, our experiments show that including the rare class does not distort overall model performance or the relative ranking of baselines.
>
> >5. No results for the imputer strategy proposed in section 3.4
>
> Response: Thank you for raising this concern. As the reviewer correctly noted, a key component of our approach is the missingness-aware **Learned Imputer (LI)** described in Section 3.4. To directly evaluate its benefit, we conducted a new experiment that compares our method with and without the LI added as Appendix E. The new experiment focuses on nodes with incomplete metadata, which is the natural setting where imputation plays an important role.
>
> In this study, we progressively masked node features at rates of 0%, 25%, 50%, 75%, and 100%. We then evaluated both versions of the model on edge classification, link prediction, and joint performance. The full results are presented in the table below.
> | Missing Features | Methods     | Acc_edge ↑ | F1_edge ↑ | Auc_edge ↑ | Acc_link ↑ | F1_link ↑ | Auc_link ↑ | Acc_joint ↑ |
> |:-----------------|:------------|-----------:|----------:|-----------:|-----------:|----------:|-----------:|------------:|
> | 0%               | Without LI  |       0.94 |      0.93 |       0.96 |       0.91 |      0.94 |       0.94 |        0.94 |
> |                  | With LI     |       0.93 |      0.92 |       0.93 |       0.91 |      0.94 |       0.93 |        0.93 |
> | 25%              | Without LI  |       0.88 |      0.87 |       0.89 |       0.90 |      0.91 |       0.93 |        0.91 |
> |                  | With LI     |       0.91 |      0.91 |       0.92 |       0.94 |      0.93 |       0.94 |        0.93 |
> | 50%              | Without LI  |       0.81 |      0.82 |       0.84 |       0.84 |      0.83 |       0.86 |        0.84 |
> |                  | With LI     |       0.87 |      0.88 |       0.90 |       0.92 |      0.91 |       0.92 |        0.90 |
> | 75%              | Without LI  |       0.71 |      0.72 |       0.75 |       0.75 |      0.74 |       0.75 |        0.73 |
> |                  | With LI     |       0.84 |      0.83 |       0.87 |       0.86 |      0.85 |       0.86 |        0.85 |
> | 100%             | Without LI  |       0.43 |      0.41 |       0.43 |       0.49 |      0.50 |       0.55 |        0.41 |
> |                  | With LI     |       0.45 |      0.41 |       0.42 |       0.47 |      0.49 |       0.52 |        0.37 |
>
> The results show a clear pattern. When no features are missing, both methods perform similarly, which is expected because imputation has no effect in this case. However, as missingness increases, the Learned Imputer consistently provides stronger performance. The improvements are especially pronounced at 50% and 75% missingness. For example, at 75% missingness, the Learned Imputer improves F1_edge from 0.72 to 0.83 and improves Acc_joint from 0.73 to 0.85. These results demonstrate that explicitly modeling missingness and providing the mask to the encoder significantly increases robustness to incomplete metadata, which is a common property of real AI supply-chain graphs.
>
> At 100% missingness, the two methods perform similarly. This is expected because neither model has access to any observed features, and the task relies primarily on structural information. This further supports our design choice: the Learned Imputer offers substantial benefits when meaningful partial information is available, and it behaves reasonably when no information is present.

---

> ### Author Response · Authors · 2025-11-25
> **Response to Reviewer iqPC [3/3]**
>
> >6. Improvements are incremental over GCN+Graphormer models in table 2 and likely statistically insignificant (no error bars are).
>
> Thank you for raising this important point. To address these concerns, we have updated Table 2 in the paper to report the mean and standard deviation across 10 runs. The updated results show that CHAININFER consistently achieves the highest performance among all joint inference models.
>
> | Category   | Model            | Acc_edge↑   | F1_edge↑    | AUC_edge↑   | Acc_link↑   | F1_link↑    | AUC_link↑   | Acc_joint↑   | Time↓      |
> |:-----------|:------------------|----------:|----------:|----------:|----------:|----------:|----------:|----------:|---------:|
> | **GNN-only** | GCN              | 0.60±0.04 | 0.62±0.03 | 0.65±0.05 | 0.68±0.03 | 0.67±0.02 | 0.71±0.03 | 0.72±0.03 | 0.08±0.02 |
> |            | GAT              | 0.42±0.02 | 0.41±0.04 | 0.59±0.03 | 0.63±0.03 | 0.62±0.02 | 0.66±0.02 | 0.70±0.03 | 0.19±0.07 |
> |            | GIN              | 0.66±0.02 | 0.68±0.03 | 0.71±0.03 | 0.73±0.02 | 0.73±0.03 | 0.76±0.02 | 0.74±0.04 | **0.06±0.02** |
> | **GT-only** | Graphormer       | 0.78±0.02 | 0.83±0.03 | 0.84±0.03 | 0.86±0.04 | 0.84±0.02 | 0.80±0.02 | 0.86±0.03 | 2.19±0.12 |
> |            | GT               | 0.76±0.02 | 0.81±0.03 | 0.83±0.03 | 0.84±0.03 | 0.84±0.03 | 0.79±0.03 | 0.84±0.03 | 1.80±0.14 |
> | **Separate** | GCN+Graphormer   | 0.79±0.04 | 0.79±0.02 | 0.85±0.04 | 0.83±0.03 | 0.86±0.02 | 0.87±0.03 | 0.84±0.03 | 2.41±0.09 |
> |            | GAT+Graphormer   | 0.80±0.02 | 0.80±0.03 | 0.83±0.04 | 0.84±0.04 | 0.85±0.03 | 0.88±0.03 | 0.86±0.02 | 2.45±0.08 |
> |            | GIN+Graphormer   | 0.81±0.04 | 0.80±0.02 | 0.85±0.03 | 0.85±0.02 | 0.86±0.03 | 0.87±0.03 | 0.89±0.03 | 2.34±0.15 |
> |            | GCN+GT           | 0.78±0.03 | 0.79±0.03 | 0.82±0.03 | 0.84±0.02 | 0.87±0.03 | 0.86±0.03 | 0.84±0.04 | 1.82±0.07 |
> |            | GAT+GT           | 0.77±0.01 | 0.82±0.03 | 0.83±0.03 | 0.83±0.02 | 0.83±0.04 | 0.86±0.03 | 0.87±0.03 | 2.01±0.09 |
> |            | GIN+GT           | 0.77±0.03 | 0.82±0.04 | 0.84±0.03 | 0.85±0.02 | 0.86±0.03 | 0.85±0.03 | 0.86±0.02 | 1.85±0.07 |
> | **Joint**   | GCN+Graphormer   | 0.89±0.03 | 0.91±0.05 | 0.92±0.02 | 0.91±0.02 | 0.92±0.01 | 0.92±0.03 | 0.91±0.02 | 3.77±0.06 |
> |            | GAT+Graphormer   | 0.88±0.04 | 0.91±0.02 | 0.93±0.02 | 0.91±0.03 | 0.92±0.04 | 0.93±0.02 | 0.90±0.02 | 4.14±0.10 |
> |            | GIN+Graphormer   | 0.90±0.04 | 0.91±0.04 | 0.92±0.03 | **0.92±0.04** | **0.94±0.03** | 0.93±0.01 | 0.91±0.02 | 3.51±0.09 |
> |            | GCN+GT           | 0.87±0.02 | 0.92±0.03 | 0.94±0.02 | 0.91±0.02 | 0.91±0.01 | 0.91±0.02 | 0.91±0.03 | 2.57±0.11 |
> |            | GAT+GT           | 0.88±0.02 | 0.92±0.02 | 0.94±0.03 | 0.90±0.02 | 0.93±0.03 | **0.95±0.01** | 0.90±0.02 | 2.80±0.11 |
> |            | **CHAININFER**   | **0.94±0.02** | **0.93±0.01** | **0.96±0.02** | 0.91±0.03 | **0.94±0.02** | 0.94±0.01 | **0.94±0.03** | 2.34±0.06 |
>
> **(1) Error bars and statistical significance.** These error bars allow a direct comparison of the variance across repeated runs. The results indicate that the improvements of CHAININFER over other baselines (e.g., GCN + Graphormer) are larger than one standard deviation for the majority of metrics. This demonstrates that the improvements are statistically meaningful rather than noise-level fluctuations.
>
> **(2) Interpretation of the variants.** All joint models evaluated in Table 2 share the same architecture as CHAININFER. The only difference lies in the choice of the GNN encoder and GT encoder. **These models should be viewed as controlled variants of the CHAININFER framework rather than unrelated baselines.** We selected the GIN + GT configuration as the primary CHAININFER model for two reasons. First, it provides the best accuracy and F1 performance for both link prediction and edge-type classification. Second, it is the most computationally efficient among the variants, which is important for real-world supply chain settings where rapid updates and large incremental graph additions are common.
>
> **(3) Significance of the performance gaps.** Based on the results, the following comparisons are statistically significant. (i) Joint models outperform non-joint models with non-overlapping confidence ranges for the key metrics. This confirms the benefit of jointly modeling link prediction and edge-type inference. (ii) Within the joint models, the performance of GIN + GT (the CHAININFER configuration) consistently exceeds that of the strongest alternative configuration. The improvements are larger than the reported variances, indicating that the gains are statistically reliable.

---

> ### Comment · Reviewer_iqPC · 2025-11-25
> **Response**
>
> Thanks for the response!
>
> There are two remaining concerns that I have that are unaddressed.
>
> 1. **On benchmarks**. The method does well on the authors' dataset but that's just a single dataset. The generalizability of the approach will be stronger if there were other benchmarks or problems where this method could be applied and good results are demonstrated. As it is, joint edge classification and link prediction (in the supply chain scenario nonetheless) feel like limited novelty to me. The dataset on its own is not that novel or challenging given only 3+1 rare edge types.
> 2. **Impute baselines**. I appreciate the new results on the impute strategy but you need to include reasonable impute baselines or tell me that there are none (not even dumb simpleimpute style ones).

---

> ### Author Response · Authors · 2025-11-26
> **Response to Reviewer iqPC [1/2]**
>
> Response: We thank the reviewer for clarifying this. We totally agree that demonstrating generalizability beyond a single dataset is necessary. Although we did not find any existing benchmark that directly targets the same provenance-oriented tasks as our paper, we identified several public graph benchmarks whose structures can be appropriately adapted to our problem formulation. Particularly, we conducted a new experiment on three public datasets as shown in the table.
>
> | Graph        | Nodes     | Edges      | NodeFeat     | Edge type | Node type                 | Graph type      |
> |--------------|-----------|------------|---------------|-------|------------------------|-----------|
> | OGB-WikiKG2  | 2.50M     | 16.11M     | None          | 535   | People, Countries,...  | Heterogeneous|
> | MovieLens-1M | 10k       | 1.00M      | Demo, Genres  | 5     | Users, Movies          | Heterogeneous|
> | SciCite      | 9,221     | 8,243      | Citation txt  | 3     | Papers                 | Homogeneous      |
>
> **New Experiment:** We compared our method against five baselines: a standard GIN [ICLR 2019], a pure Graph Transformer (GT) [AAAI 2021], a pure Graphormer [NeurIPS 2021] and two separately trained models. The following table summarizes the results.
>
> **(1) SciCite (Citation Intent)**
>
> SciCite presents a challenging, imbalanced, semantically rich edge-classification task. As shown in the table, ChainInfer achieves the highest Acc_edge (0.69), F1_edge (0.61), and AUC_edge (0.73), outperforming all baselines by 8 – 18 points. More importantly, the joint performance, measured by Acc_joint, improves from 0.53 (best separate baseline) to 0.72. This demonstrates that modeling the dependency between link formation and edge semantics is beneficial even when edge labels represent complex linguistic intent.
>
> **(2) OGB-WikiKG2 (535 relation types)**
>
> WikiKG2 is an extremely large and highly multi-relational knowledge graph containing 535 edge types, a lot more than the 4 edge types in our original dataset. Despite the much larger and more challenging label space, ChainInfer achieves the best performance across all metrics, including F1_edge = 0.66, AUC_edge = 0.81, and Acc_joint = 0.80, a 4 - 8 point gain over the strongest baseline. The improvements on such a large relation vocabulary confirm that the proposed method scales to significantly more complex multi-class edge semantics.
>
> **(3) MovieLens (User-Item, 1-5 rating prediction)**
>
> MovieLens provides a fine-grained, ordinal edge classification setting. Here again ChainInfer obtains the top performance (e.g., F1_edge = 0.76, AUC_edge = 0.78, Acc_joint = 0.80), improving over the next best model by 3 - 6 points. The consistent gains across both edge-type and link-level metrics indicate that the joint modeling benefit is not limited to provenance-style edges but generalizes to transactional and bipartite graph domains.
>
> | Data | Model      | Acc_edge | F1_edge | AUC_edge | Acc_link | F1_link | AUC_link | Acc_joint |
> |------|------------|-------|------|-------|-------|------|--------|--------|
> | **SciCite** | GIN | 0.35 | 0.47 | 0.61 | 0.51 | 0.67 | 0.58 | 0.58 |
> |      | GT         | 0.41 | 0.44 | 0.61 | 0.51 | 0.61 | 0.49 | 0.49 |
> |      | Graphormer | 0.51 | 0.54 | 0.68 | 0.58 | 0.55 | 0.52 | 0.52 |
> |      | Sep GIN+Graphormer | 0.61 | 0.58 | 0.68 | 0.61 | 0.66 | 0.53 | 0.53 |
> |      | Sep GIN+GT   | 0.61 | 0.58 | 0.67 | 0.64 | 0.63 | 0.57 | 0.57 |
> |      | ChainInfer | **0.69** | **0.81** | **0.73** | **0.83** | **0.83** | **0.79** | **0.72** |
> | **WikiKG2** | GIN | 0.51 | 0.43 | 0.55 | 0.67 | 0.67 | **0.88** | 0.67 |
> |      | GT         | **0.81** | 0.51 | 0.61 | 0.68 | 0.71 | 0.71 | 0.71 |
> |      | Graphormer | 0.79 | 0.53 | 0.63 | 0.73 | 0.70 | 0.81 | 0.78 |
> |      | Sep GIN+Graphormer | 0.77 | 0.53 | 0.72 | 0.74 | 0.74 | 0.78 | 0.72 |
> |      | Sep GIN+GT   | 0.77 | 0.55 | 0.74 | 0.78 | 0.71 | 0.77 | 0.77 |
> |      | ChainInfer | 0.79 | **0.66** | **0.81** | **0.82** | **0.83** | 0.85 | **0.79** |
> | **MovieLens** | GIN | 0.66 | 0.64 | 0.66 | 0.75 | 0.76 | 0.81 | 0.71 |
> |      | GT         | 0.75 | 0.70 | 0.71 | 0.81 | 0.82 | 0.79 | 0.72 |
> |      | Graphormer | **0.78** | 0.75 | **0.81** | 0.74 | 0.76 | 0.80 | 0.79 |
> |      | Sep GIN+Graphormer | 0.72 | 0.68 | 0.72 | 0.74 | 0.72 | 0.75 | 0.75 |
> |      | Sep GIN+GT   | 0.76 | 0.68 | 0.77 | **0.82** | 0.79 | 0.74 | 0.79 |
> |      | ChainInfer | 0.77 | **0.76** | 0.78 | 0.78 | **0.84** | **0.84** | **0.80** |

---

> ### Author Response · Authors · 2025-11-26
> **Response to Reviewer iqPC [2/2]**
>
> >Impute baselines. I appreciate the new results on the impute strategy but you need to include reasonable impute baselines or tell me that there are none (not even dumb simpleimpute style ones).
>
> Response:
>
> We thank the reviewer for clarifying this. We conducted a new experiment to add three imputation baselines:
>
> - **Mean Imputation:** A standard univariate baseline.
>
> - **MICE (Multivariate Imputation by Chained Equations):** A widely used statistical method that models feature correlations.
>
> - **k-NN Imputation:** A geometric baseline that utilizes local neighborhood structure.
>
> The results, summarized in Table below, demonstrate that our end-to-end Learnable Imputer consistently outperforms or matches these baselines, particularly in realistic missingness scenarios (25% - 75%).
>
> First, we can see at 25% missingness, our method achieves a Joint Accuracy of 0.93, significantly outperforming other pre-processing baselines. This indicates that learning imputation jointly with the downstream task is more effective than pre-processing.
>
> Second, as data becomes scarcer, the gap widens. At 75% missingness, the "No Imputer" baseline drops to 0.73, and Mean Imputation drops to 0.77. Our method maintains a robust 0.85 Joint Accuracy, demonstrating its ability to leverage graph topology when feature signals are weak.
>
> Third, while k-NN performs competitively at 50% missingness, it scales quadratically ($O(N^2)$), making it computationally prohibitive for large graphs. Our Learnable Imputer achieves comparable or superior accuracy but only with linear complexity ($O(N)$), making it the optimal choice for large-scale supply chain graphs.
>
> | Missing Features | Imputer Method | Acc_edge | F1_edge | Auc_edge | Acc_link | F1_link | Auc_link | Acc_joint |
> |------------------|----------------|----------|---------|----------|----------|---------|----------|-----------|
> | **0%**   | No imputer     | **0.94**     | 0.93    | **0.96**     | 0.91     | **0.94**    | 0.94     | 0.94      |
> |                  | Mean imputer   | 0.92     | 0.93    | 0.94     | 0.92     | 0.93    | **0.95**     | **0.95**   |
> |                  | MICE imputer   | 0.93     | 0.92    | 0.94     | **0.93**    | 0.92    | 0.94     | 0.91      |
> |                  | KNN imputer    | 0.92     | **0.94**    | 0.94     | 0.92     | 0.92    | 0.93     | 0.94      |
> |                  | Ours           | 0.93     | 0.92    | 0.94     | 0.91     | **0.94**    | 0.93     | 0.93      |
> | **25%**  | No imputer     | 0.88     | 0.87    | 0.89     | 0.90     | 0.91    | 0.93     | 0.91      |
> |                  | Mean imputer   | 0.89     | **0.91**    | 0.91     | 0.83     | 0.81    | 0.86     | 0.89      |
> |                  | MICE imputer   | **0.91**     | 0.89    | **0.92**     | 0.85     | 0.82    | 0.86     | 0.89      |
> |                  | KNN imputer    | 0.81     | 0.83    | 0.86     | 0.92     | 0.89    | 0.92     | 0.87      |
> |                  | Ours           | **0.91**     | **0.91**    | **0.92**     | **0.94**     | **0.93**    | **0.94**     | **0.93**      |
> | **50%**  | No imputer     | 0.81     | 0.82    | 0.84     | 0.84     | 0.83    | 0.86     | 0.84      |
> |                  | Mean imputer   | 0.85     | 0.87    | 0.89     | 0.84     | 0.86    | 0.88     | 0.86      |
> |                  | MICE imputer   | **0.87**     | **0.89**    | **0.91**     | 0.88     | 0.91    | 0.91     | 0.89      |
> |                  | KNN imputer  | 0.85    | 0.85  | 0.87     | **0.92**     | **0.93**    | **0.95**     | **0.90**      |
> |                  | Ours           | **0.87**     | 0.88    | 0.90     | **0.92**     | 0.91    | 0.92     | **0.90**      |
> | **75%**  | No imputer     | 0.71     | 0.72    | 0.75     | 0.75     | 0.74    | 0.75     | 0.73      |
> |                  | Mean imputer   | 0.74     | 0.77    | 0.86     | 0.83     | 0.81    | **0.88**     | 0.77      |
> |                  | MICE imputer   | 0.81     | **0.84**    | 0.85     | **0.89**     | 0.79    | 0.82     | 0.83      |
> |                  | KNN imputer    | 0.77     | 0.74    | 0.83     | 0.85     | 0.82    | 0.86     | 0.80      |
> |                  | Ours           | **0.84**     | 0.83    | **0.87**     | 0.86     | **0.85**    | 0.86     | **0.85**      |
> | **100%** | No imputer     | 0.43     | 0.41    | 0.43     | 0.49     | 0.50    | 0.55     | 0.41      |
> |                  | Mean imputer   | **0.45**     | 0.41    | 0.42     | 0.47     | 0.49    | 0.52     | 0.37      |
> |                  | MICE imputer   | 0.42     | 0.38    | **0.49**     | 0.47     | 0.49    | 0.52     | 0.37      |
> |                  | KNN imputer    | 0.41     | **0.43**    | 0.45     | **0.51**     | **0.53**    | **0.59**     | **0.43**      |
> |                  | Ours           | **0.45**     | 0.41    | 0.42     | 0.47     | 0.49    | 0.52     | 0.37      |

---

> > ### Comment · Reviewer_iqPC · 2025-11-27
> > **Thanks for the rapid follow up! My concerns are addressed and I have increased the score**
> >
> > Thanks for the rapid follow up!

---

> > > ### Author Response · Authors · 2025-12-01
> > > **Response to reviewer iqPC**
> > >
> > > We sincerely thank you for your continued engagement and for confirming that our revisions have successfully addressed your concerns. We are deeply grateful for your re-evaluation and for raising your score from 2 to 6. Your constructive feedback has significantly helped us improve the quality and robustness of our work.

---

### Author Response · Authors · 2025-11-25
**Revised Paper Uploaded**

We thank all reviewers for their valuable comments and feedback. We have made a concerted effort to address every concern and answer all questions below. In addition, we have revised and uploaded our paper accordingly to reflect the changes, which are marked in $\color{blue}blue$.

---

### Author Response · Authors · 2025-12-01
**General response**

We sincerely thank all reviewers for their constructive feedback and for acknowledging the novelty of our problem formalization and proposed solution. We are encouraged that the reviewers have acknowledged that we have addressed all their concerns.

Below, we summarize our major updates, specifically highlighting six new experiments and key textual clarifications.

1. New Experiments and Ablations. To address concerns regarding robustness, generalizability, and baselines, we have added six new experiments in the Appendix:

    - **Effectiveness of Imputer Strategy** (Appendix E): Evaluated the learned imputer to address Reviewer iqPC (Q5) and Reviewer 6s7p (Q1).

    - **Generalization Capabilities** (Appendix F): Demonstrated generalization to “dataset” entities, addressing Reviewer KggE (Q4) and Reviewer 6s7p (Q3).

    - **Feature Ablation** (Appendix G): Verified the contribution of specific feature categories as requested by Reviewer 6s7p (Q4).

    - **Robustness Analysis** (Appendix H): Tested model performance under noisy/incorrect inputs to confirm stability, addressing Reviewer 6s7p (Q2).

    - **Comparison with Non-Graph Baselines** (Appendix I): Added comparisons to demonstrate the superiority of our graph-based approach over non-graph methods, addressing Reviewer KggE (Q2) and Reviewer 6s7p (Q5).

    - **Explainability** (Appendix J): Provided qualitative examples to improve interpretability, addressing Reviewer 6s7p (Q6).

2. Clarifications and Textual Revisions

    - **Novelty & Positioning:** Following the discussion with Reviewer iqPC, we have revised the text to better clarify the novelty of our model and dataset compared to existing work (Addressing Reviewer iqPC Q1, Q4).

    - **Graph Construction & Hardware:** We provided a detailed, step-by-step explanation of the graph construction process. Additionally, we detailed the hardware configuration in Section 4.1 and modified the result tables for better clarity (Addressing Reviewer 7JT6 Q1 & other suggestions).

    - **Scope, Related Work & Metrics:** We refined the scope definition in the Introduction. Additionally, we expanded the discussion on related work regarding missing metadata inference in other domains. Furthermore, we added detailed definitions and justifications for our evaluation metrics (Addressing Reviewer KggE Q1, Q3).

---

### Note · Program_Chairs · 2026-01-17
**Submission Desk Rejected by Program Chairs**

The following references in this submission do not refer to real documents and/or have major errors in bibliographic information:

 NIST. Artificial intelligence supply chain risk management framework. Technical report, U.S. Department of Commerce, 2023b. NIST AI RMF Supplementary Guidance.
Tianyi Zhu and Thomas Zimmermann. An empirical study on the build failures in the npm ecosystem. In MSR, 2018.
Linyong Zhang, Haixu Tang, Yizhou Zhao, and Muhao Li. Extracting secrets from chatgpt: Towards measuring the resilience of language models against extraction attacks. arXiv preprint arXiv:2307.01952, 2023.
Andy Chen, Xinyang Liu, Dawn Song, et al. Jailbreak chat: A benchmark for jailbreaking large language models, 2023a.
Abeba Birhane, Carina Prunkl, Caroline Kaeser-Chen, Leo A. König, Amandalynne Paullada, Laura Strohm, and Briana Vecchione. The scientific competence gap in large language models. NeurIPS 2023 Workshop on Navigating and Addressing Data Problems for Foundation Models, 2023. URL https://arxiv.org/abs/2308.05374.
Weijia Shi, Qian Song, Zhenyu Liang, et al. Unalignment: An empirical study of misalignment in aligned language models. In $I C L R, 2024$.